# THE EXPRESSIVE POWER OF $k$-HARMONIC DISTANCES FOR MESSAGE PASSING NEURAL NETWORKS

## ABSTRACT

Positional encodings from spectral graph theory—such as spectral distances like effective resistance—have been shown to enhance the performance of graph neural networks (GNNs). However, the theoretical expressive power of these spectral features is not entirely understood. While certain spectral features are known to increase expressive power, it is unclear if different spectral features are equally powerful. Moreover, while it is established that spectral distance measures can enhance the expressivity of transformer-based architectures, their implications for message passing neural networks (MPNNs) are relatively underexplored. In this work, we focus on one such family of spectral features: the $k$-harmonic distances. We establish upper and lower bounds on the expressivity of MPNNs augmented with $k$-harmonic distances. We also show that not all $k$ are equally expressive, and some are better than others in certain situations. To corroborate this theory, we present several empirical results demonstrating $k$-harmonic distance's expressive power. We show its potential for computational efficiency over transformers in some cases. Further, we experiment with making $k$ a learnable parameter and find that different datasets have different optimal values of $k$.

## 1 INTRODUCTION

Graph neural networks (GNNs) have been extensively studied in the machine learning community. GNNs have demonstrated strong performance on a variety of tasks, including node classification, graph classification, and regression. However, GNNs are fundamentally limited in their ability to capture global structural information, which is essential for many graph learning problems.

This limitation has been formalized (Xu et al., 2018; Morris et al., 2019) by showing that message-passing neural networks (MPNNs), perhaps the dominant class of GNNs, are no more powerful than the Weisfeiler-Lehman (WL) graph isomorphism test (Weisfeiler & Lehman, 1968). In practice, MPNNs are further limited by issues such as oversmoothing (Oono & Suzuki, 2020; Cai & Wang, 2020) which harm the performance of MPNNs as the number of layers increase.

*Positional encodings* (PEs) have been proposed as a solution to both limitations. They can provide the missing global structural information in the form of node or edge features, and can curb the issue of oversmoothing by requiring fewer layers of message passing for this global structural information to propagate through the graph, resulting in a *shallow GNN* (i.e. a GNN with few layers.)

Various positional encodings have been explored to provide different types of structural information to GNNs, including shortest-path distances (Ying et al., 2021) and spectral distances such as effective resistance (a.k.a. commute time) (Zhang et al., 2023; Velingker et al., 2023). However, the broader question of which positional encoding is best suited for any given task remains open—both in theory (expressive power) and in practice (empirical performance).

A broad class of positional encodings are derived from the eigenvalues and eigenvectors of the graph Laplacian, so-called *spectral positional encodings*. Some of these encodings are *relative* (they assign values to pairs of nodes), while others are *absolute* (they assign values to individual nodes). A GNN that makes use of spectral PEs is said to be a *spectral GNN*. Further, a major class of relative spectral PEs are spectral distances, defined generally as

$$D_f(s,t) = \sqrt{(1_s - 1_t)^\top f(L)(1_s - 1_t)}, \tag{1}$$

where $f(L)$ is a matrix function of the Laplacian and $1_x$ is an indicator vector that is 1 at index $x$ and zero everywhere else. Well-known examples of such distances include *effective resistance*[1] (Kirch-hoff, 1847), where $f(L) = L^+$, *biharmonic distance* (Lipman et al., 2010), where $f(L) = (L^+)^2$, *$k$-harmonic distance* (Black et al., 2024a), where $f(L) = (L^+)^k$, and *diffusion distance* at time $t$ (Coifman & Lafon, 2006), where $f(L) = e^{-tL}$.[2]

The goal of this work is to advance our understanding of the expressive power of spectral positional encodings. To this end, we show that the class of $k$-harmonic distances is as expressive as any other spectral distance (Theorem 6.4). This result allows us to focus on analyzing the expressive power of $k$-harmonics as positional encodings, with implications for the broad class of spectral encodings.

Different $k$-harmonic distances are known to encode several fundamental properties of a graph, making them a good candidate for use as a PE. The effective resistance between nodes $s$ and $t$ is lower the more different paths exist between $s$ and $t$ so it measures how *well-connected* $s$ and $t$ are. It is also connected to the expected number of steps in random walks between $s$ and $t$ (Chandra et al., 1996). The biharmonic distance has been shown to measure how important or central an edge $(s,t)$ is to the overall structure of a graph (Li & Zhang, 2018; Yi et al., 2018a; Black et al., 2024a).

While there are known upper bounds on the expressivity for generic spectral positional encodings that subsume the spectral distances (e.g., the eigenspace projection invariant (Zhang et al., 2024; Gai et al., 2025)), these positional encodings rely on projections into Laplacian eigenspaces which are not computationally feasible for most graph data, warranting an exploration of more efficient, but potentially less expressive, positional encodings like the $k$-harmonic distances. Moreover, with the exception of (Velingker et al., 2023) and (Feldman et al., 2023), most of the theoretical work on spectral positional encodings have been for transformers (Zhang et al., 2023) or fully-connected GNNs like IGNs (Black et al., 2024b; Zhang et al., 2024) whose runtime scales quadratically with the number of nodes, in contrast to MPNNs that scale linearly with the number of edges.

Despite their generality, the power of $k$-harmonic distances as positional encodings for GNNs is not well understood beyond the case of $k = 1$ (Velingker et al., 2023; Zhang et al., 2023). Many questions remain open: What is the expressive power of different values of $k$? Are some values more informative for specific tasks or graph structures? Can combining multiple $k$-harmonics yield more powerful representations? And how effective are these encodings in practice, particularly in shallow GNNs with limited receptive fields?

**Contributions** We address these questions through theoretical analysis and empirical evaluation:

- We show that the set of the first $2n$ $k$-harmonic distances for an $n$-vertex graph is at least as expressive as any other spectral distance for MPNNs (Theorem 6.4).

- We show that $k$-harmonic positional encodings strictly increase the expressive power of MPNNs beyond the WL test (Theorem 5.1), but not beyond the 3-WL test (Theorem 5.2).

- As Theorem 6.1 shows that different $k$-harmonic different information about the graph, this suggests combining multiple $k$-harmonics increases expressivity. However, this benefit has diminishing returns: we prove that for any graph, the number of distinct $k$-harmonics that contribute new information is at most the number of distinct Laplacian eigenvalues (Theorem 6.3). At this point, the concatenated $k$-harmonics are as powerful as general spectral positional distance (Theorem 6.4).

- We show that different $k$-harmonics capture different structural properties in shallow GNNs (Theorem 6.1), and thus, there are better (more expressive) values of $k$ for different tasks and types of graphs. Conversely, we show that if a pair of graphs can be distinguished by some $k$-harmonic, then they can be distinguished for all but finitely many other $k$-harmonic distances (Theorem 6.2).

---

[1]Technically, effective resistance is defined $R(s,t) = (1_s - 1_t)^\top L^+ (1_s - 1_t)$, i.e., with no square root. Surprisingly, even without the square root, effective resistance is a metric (Klein & Randić, 1993, Theorem B)

[2]$L^+$ denotes the Moore-Penrose pseudoinverse of the Laplacian $L$.

- We empirically validate (Theorem 5.1) and (Theorem 5.2) using the BREC dataset (Wang & Zhang, 2024) as a measure of realized expressivity. To validate the differences between individual $k$, we evaluate on ZINC (Dwivedi et al., 2023) and ogbg-molhiv (Hu et al., 2020a), confirming that the optimal choice of $k$ depends on the dataset or task at hand. This motivates using multiple or learnable $k$ values in practice. We also implement a learnable-$k$ parameter and report its performance.

- Finally, we observe that $k$-harmonic encodings can improve training efficiency. On ZINC, $k$-harmonic MPNNs offer a favorable trade-off between training time and accuracy compared to more expensive graph transformers.

## 2 RELATED WORK

**$k$-harmonic distances** Though the $k$-harmonic distance is a natural extension of the effective resistance and biharmonic distance, it was only recently proposed (Black et al., 2024a) and has not been fully explored. Its efficacy has been shown for clustering, but little else is known.

The effective resistance is much more well-studied in the literature. It has been shown several times that the effective resistance of a graph is a measure of connectivity and has been utilized for various graph-specific tasks such as sparsification (Spielman & Srivastava, 2011), clustering (Alev et al., 2018), and is now being used as an positional encoding in GNNs (Velingker et al., 2023; Zhang et al., 2023). As it relates to our work, it has been shown that MPNNs that make use of effective resistance are strictly more expressive than the WL test (Velingker et al., 2023).

The biharmonic distance has been used in the study of consensus networks (Yi et al., 2018b; 2021) and has been shown to be a measure of edge centrality (Li & Zhang, 2018; Yi et al., 2018a; Black et al., 2024a). However, it has not been studied as a positional encoding in GNNs.

**Positional Encodings from Spectral Graph Theory** While we consider incorporating spectral information into GNNs using the $k$-harmonic distances, there have been many previously proposed methods for incorporating spectral information in other ways. The first graph transformers used Laplacian eigenvectors as positional encodings (Dwivedi & Bresson, 2021), with subsequent works also using Laplacian eigenvectors as positional encodings (Kreuzer et al., 2021; Rampasek et al., 2022; Zhou et al., 2024). However, Laplacian eigenvectors suffers from sign and basis ambiguities, so Lim et al. (2023) proposed to use the projection onto the eigenspaces, rather than the eigenvectors themself, to avoid this ambiguity. Huang et al. (2024) and Zhang et al. (2024) proposed alternative techniques using the projections onto the eigenspaces. Other graph neural networks have used spectral invariants beyond the eigenvectors as positional encodings, including the effective resistance (Zhang et al., 2023; Velingker et al., 2023) and heat kernels (Choromanski et al., 2022; Feldman et al., 2023).

**Expressivity of Spectral Invariants for Graph Isomorphism** Understanding the capability of spectral invariants, such as spectral distances, to distinguish non-isormorphic graphs has recently been an active area of research as its relate to graph learning, specifically in the area of graph transformer. However, understanding which pairs of graphs are distinguished by different spectral invariants predates GNNs. Fürer (2010) proposed a spectral invariant that he showed was weaker than the 3-WL test. Rattan & Seppelt (2023) showed Furer's invariant was *strictly* weaker than the 3-WL test and the (1,1)-WL test. Zhang et al. (2023) studied the expressive power of effective resistance as a relative positional encoding for a transformer. They proposed the RD-WL test—a variant of the WL test that incorporates effective resistance—as an upper bound of the expressive power of these transformers and showed this was weaker than the 3-WL test. The resistance-distance transformer was generalized to eigenspace projection neural networks (EPNNs) by Zhang et al. (2024). They also introduced the eigenspace projection WL (EP-WL) test as an upper bound for EPNNs, which they proved was weaker than both 3-WL and certain types of subgraph WL tests.

## 3 BACKGROUND

Let $G = (V, E)$ be an undirected, unweighted graph. Denote the number of vertices and edges as $n = |V|$ and $m = |E|$. Additionally, let the graph have a set of node features, $\{x_v \in \mathbb{R}^d : v \in V\}$,

and a set of edge features, $\{e_{uv} \in \mathbb{R}^f : (u,v) \in E\}$. The **adjacency matrix** of $G$ is the matrix $A \in \mathbb{R}^{n \times n}$ where $A_{i,j} = 1$ if $(i,j) \in E$ and 0 otherwise. The **degree matrix** is the diagonal matrix $D \in \mathbb{R}^{n \times n}$ where $D_{i,i} = \deg(i)$. The **Laplacian matrix** is $L = D - A$ and is the central structure of study in spectral graph theory, and is positive semidefinite by definition. The eigenvalues of the Laplacian are the **spectrum** of the graph. Throughout this paper, multisets are denoted with the double curly bracket notation $\{\!\{\}\!\}$.

**Message Passing Neural Networks**  **Graph neural networks** are functions that take as input a graph $G = (V, E)$, a set of node features, $\{x_v \in \mathbb{R}^d : v \in V\}$, and (optionally), a set of edges $\{e_{uv} \in \mathbb{R}^f : (u,v) \in E\}$. The $t^{\text{th}}$ layer of a GNN computes features $h_v^{(t)}$ for each vertex using the node features from the previous layer $h_v^{(t-1)}$. The most common type of graph neural network are **message passing neural network (MPNNs)** (Gilmer et al., 2017). Each layer of an MPNN updates the feature of a node by aggregates the feature from its neighbors $h_u^{(t)}$ and the features of the incident edegs $e_{uv}$. Initially, $h_v^{(0)} = x_v$. For each layer $t \in \{1, ..., T\}$, a message passing layer updates the node features using the following formula:

$$h_v^{(t)} = \phi^{(t)} \left( h_v^{(t-1)}, \psi^{(t)} \left( \{\!\{ (e_{uv}, h_u^{(t-1)}) : (u,v) \in E \}\!\} \right) \right), \tag{2}$$

where $\phi^{(t)}$ and $\psi^{(t)}$ are learnable functions and $\psi^{(t)}$ is invariant on multisets, e.g. sum.

**WL Tests**  The **Weisfeiler-Lehman (WL) test** is an iterative algorithm that assigns labels to nodes in order to deduce whether or not two graphs are isomorphic. Specifically, the WL test assigns each vertex $v \in V$ a color $\chi^{(t)}(v)$ for all $t \geq 0$. The labels are initialized to some arbitrary constant in the $0^{\text{th}}$ iteration, e.g., $\chi^{(0)}(v) = 1$ for all $v \in V$. For $t \geq 1$, the WL color of a vertex $v$ is defined

$$\chi^{(t)}(v) = \text{hash} \left( \chi^{(t-1)}(v), \{\!\{ \chi^{(t)}(u) : (u,v) \in E \}\!\} \right) \tag{3}$$

where $\text{hash}$ is an injective hash function.

Let $\chi^{(t)}(G) = \{\!\{ \chi^{(t)}(v) : v \in V_G \}\!\}$. Two graphs $G$ and $H$ are **indistinguishable** by the WL test if they have the multisets of colors for all $t \geq 0$, or formally,

$$\chi^{(t)}(G) = \{\!\{ \chi^{(t)}(v) : v \in V_G \}\!\} = \{\!\{ \chi^{(t)}(v) : v \in V_H \}\!\} = \chi^{(t)}(H) \qquad \forall t \geq 0.$$

$G$ and $H$ are **indistinguishable by $T$ iterations** of the WL test if $\chi^{(t)}(G) = \chi^{(t)}(H)$ for $T \geq t \geq 0$.

The WL test provides an upper bound on the distinguishing power of MPNNs with constant node features  (Xu et al., 2018; Morris et al., 2019).

Further, there are higher order variants of the WL test called the **k-WL tests** that assign colors to tuples of $k$ nodes rather than to single nodes; see (Huang & Villar, 2021).

**Comparing Isomorphism Tests**  The WL test is a **one-sided graph isomorphism test**. That is, if two graphs are distinguishable by the WL test they are guaranteed to be non-isomorphic, but two graphs that are indistinguishable are not guaranteed to be isomorphic. MPNNs are also a one-sided graph isomorphism test. An MPNN distinguishes a pair of graphs $G$ and $H$ if the multiset of node features are different, i.e., $\{\!\{ h_v^{(t)} : v \in V_G \}\!\} \neq \{\!\{ h_v^{(t)} : v \in V_H \}\!\}$.

We are interested in comparing isomorphism tests. Suppose we have two different one-sided isomorphism tests, $A$ and $B$, that for any two graphs $G$ and $H$ return if they are distinguishable or indistinguishable. Test $A$ is said to be **as strong as** test $B$ if any two graphs $G$ and $H$ that are indistinguishable by $A$ are also indistinguishable by $B$. That is, if $A$ fails to distinguish a pair of graphs, then $B$ will as well. Likewise, $A$ is **strictly stronger** than $B$ if $A$ is as strong as and there exist some pair of graphs $G$ and $H$ such that $A$ distinguishes but $B$ does not distinguish.

## 4  SPARSE $\psi$ WL TEST

In this section, we present the *sparse $\psi$ WL test*, a modification of the WL test that incorporates edge features. While the WL test provides an upper bound on the expressive power of MPNNs, the sparse

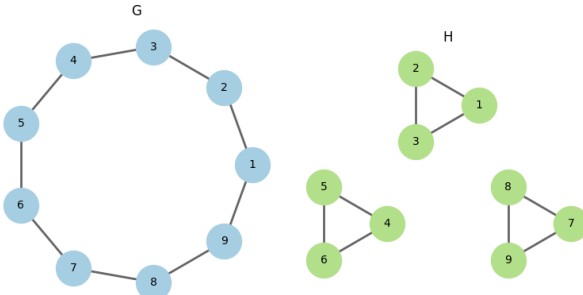

Figure 1: The graphs $G$ (the length 9 cycle graph) and $H$ (3 copies of the length 3 cycle graph) are indistinguishable by the WL test, but distinguishable by any sparse $k$-harmonic WL test

$\psi$ WL test upper bounds MPNNs *that use edge features*. In Section 5, we present several results about the expressivity of this sparse $\psi$ WL test when $\psi$ is a $k$-harmonic distance.

An ***edge positional encoding*** is a function $\psi$ that assigns each graph $G$ a map $\psi_G : E_G \to \mathbb{R}^m$ such that, if $\sigma : E_G \to E_H$ is the map on edges induced by a graph isomorphism, then $\psi_G = \psi_H \circ \sigma$; edge positional encodings complement previously defined *absolute* and *relative positional encodings* which are functions $\alpha_G : V_G \to \mathbb{R}^d$ and $\rho_G : V_G \times V_G \to \mathbb{R}^d$ respectively; see (Black et al., 2024b). In the current work, we usually take $\psi$ to be a $k$-harmonic distance, i.e., $\psi((u,v)) = H_k(u,v) \in \mathbb{R}$.

Like the WL test, the ***sparse $\psi$ WL test*** iteratively assigns labels to the vertices of a graph. The labels for the $t^{\text{th}}$ iteration are computed using the following formula:

$$\chi_\psi^{(t)}(v) = \text{hash}\left(\chi_\psi^{(t-1)}(v), \{\!\{(\psi((u,v)), \chi_\psi^{(t-1)}(u)) : (u,v) \in E\}\!\}\right) \tag{4}$$

***Indistinguishability*** for the sparse $\psi$ WL test is defined the same as for the WL test in Section 3.

It is an open question as to how different choices of $\psi$ affect the distinguishing power of the test. When $\psi$ is a constant function, the sparse $\psi$ WL is equally strong as the original WL test. Therefore, for any $\psi$, the sparse $\psi$ WL is at least as strong as the original WL test. In Section 5, we will prove that with certain choice of $\psi$, that the sparse $\psi$ WL test is *strictly* stronger than the WL test.

The reason we introduce the sparse $\psi$ WL is because it provides an upper bound on the distinguishing power of MPNNs with edge features $\psi$, as evidenced by the following lemma.

**Lemma 4.1.** *Let $G$ and $H$ be graphs with constant node features and edge features given by $\psi$. If $G$ and $H$ are indistinguishable by $t$ iterations of the sparse $\psi$ WL test, then for any $t$-layer MPNN $f$, the multisets of node features at each layer are equal, i.e., $\{\!\{h_v^{(t)} : v \in V_G\}\!\} = \{\!\{h_v^{(t)} : v \in V_H\}\!\}$*

**Related Work.** While our sparse WL test is defined for edge positional encodings, there are similar WL tests defined for relative positional encodings (RPEs) (Zhang et al., 2023; Black et al., 2024b; Zhang et al., 2024; Arvind et al., 2024; Gai et al., 2025). The RPE WL test aggregates over all nodes in an iteration; in contrast, our sparse WL test only aggregates a node's neighbors in a given iteration, hence the name "sparse." While both tests generalize the classical WL tests by incorporating positional encodings, these two WL tests play different roles in the study of GNNs. RPE WL tests provide an upper bound on the expressive power of graph transformers and similar architectures like IGNs, while our sparse WL test provides an upper bound on MPNNs with edge features.

## 5 COMPARING THE SPARSE $k$-HARMONIC AND CLASSICAL WL TESTS

In this section, we compare the sparse $k$-harmonic WL tests to classical WL tests. We show that it lies in the space between the WL test and the 3-WL test.

We show that any $k \geq 1$ yields a sparse $k$-harmonic WL test that is strictly more expressive than the WL test. This had previously been shown for $k = 1$ (i.e., effective resistance) (Velingker et al., 2023, Theorem 3.7), but it was not known for general $k$.

**Theorem 5.1.** *Let $k \geq 1$. The sparse $k$-harmonic WL is strictly stronger than the WL test.*

We defer the full proof to the Appendix, but we utilize the pair of regular graphs found in Figure 1. It is well known that the WL test is unable to distinguish pairs of regular graphs. We show that the edge features decided by any $k \geq 1$ is able to distinguish a given edge in these graphs.

**Theorem 5.2.** *The 3-WL test is strictly stronger than the sparse $k$-harmonic WL test for all $k \in \mathbb{R}$.*

In the opposite direction, we show that 3-WL is an upper bound for our sparse $k$-harmonic WL test. The proof is implied by (Zhang et al., 2024, Lemma A.16), but we include a complete proof in the Appendix.

# 6 Comparing Different $k$-Harmonic Distances

Section 5 laid the groundwork for study of sparse $k$-harmonic WL test. We now provide theoretical results for comparing different values of $k$ and comparing the $k$-harmonic distances to other spectral distances.

Per Lemma 4.1, a $t$-layer GNN is equivalent to a WL test with $t$ iterations. In this section, we provide results to show that not all values of $k$ are equally powerful. To prove this, we show that there are graphs that sparse biharmonic WL can distinguish in fewer iterations than sparse resistance WL, corresponding to a GNN with fewer layers.

In this theorem, the sparse resistance WL test and sparse biharmonic WL test to refer to the sparse $\psi$ WL test when $\psi$ is the effective resistance of biharmonic distance respectively.

**Theorem 6.1.** *There are pairs of graphs that sparse biharmonic distance WL can distinguish in one iteration but the sparse resistance WL test cannot distinguish in $o(n)$ iterations.*

The pairs of graphs we consider in this proof are both trees. To prove this theorem, we will use the following fact.

**Lemma 6.1.** *The sparse resistance WL test is equally strong as the WL test when $G$ and $H$ are trees*

This result is directly implied by a result of Ghosh et al. (2008, Theorem 2.3) that the effective resistance between any two nodes in a tree is their shortest path distance. Thus, the effective resistance of all edges in a tree is 1, giving no additional information. Conversely, Black et al. (2024a, Theorem 5.1) proves that the biharmonic distance for any edge in a tree uniquely determined by the number of nodes on either side of that cut edge, which does supply more additional topological information about an edge. The full proof of Theorem 6.1 is contained in Appendix B.1. While this may suggest that sparse biharmonic WL is much more powerful than sparse resistance WL on trees, this does not generalize to all trees. We can construct a counterexample consisting of a pair of non-isomorphic trees that sparse biharmonic WL cannot distinguish in $o(n)$ iterations. We defer this example to the appendix as well.

Importantly, this result gives credence to the idea that there is an important amount of expressive granularity that exists within the $k$-harmonics and spectral distances as a whole. That is, while all $k$-harmonics are situated between 1-WL and 3-WL, the WL hierarchy does not tell the whole story of their expressive power, and there can be stronger and weaker spectral encodings in this space.

Theorem 6.1 proves that sparse biharmonic WL can distinguish a pair of graphs much more quickly than sparse resistance WL. A natural follow-up question is if this is common. Our next theorem suggests this is not usually the case for different $k$-harmonic distances. Intuitively, if two graphs are distinguishable by some $k$-harmonic WL test, they will be distinguishable for *most* $k$-harmonic WL tests (disregarding the number of iterations $t$). A proof can be found is found in Appendix B.3.

**Theorem 6.2.** *Let $G$ and $H$ be graphs with $n$ vertices that are distinguishable by sparse $k$-harmonic WL for some $k$. Then for all but $O(n^5)$ values of $k' \in \mathbb{R}^+$, $G$ and $H$ are distinguishable by the sparse $k'$-harmonic WL test.*

We have shown that most $k$-harmonics provide equally powerful WL tests. A potential way to improve the expressive power of the $k$-harmonic distance would be to use the concatenation of *multiple* $k$-harmonic distances as an edge feature. Concatenating multiple positional encodings is a common technique that is known to increase the expressive power of a network (Ma et al., 2023; Zhang et al., 2023). However, we show that this technique has a limit. Specifically, we show that concatenating the first $2n$ $k$-harmonic distances is as strong as taking all $k$-harmonic distances. This is analogous to known results for other RPEs like powers of the adjacency matrix or heat kernels (Black et al., 2024b; Gai et al., 2025).

For a set of values $S \subset \mathbb{R}$, let the sparse $S$-harmonic WL test denote the sparse $\psi$-WL test for $\psi : E \to \mathbb{R}^{|S|}$ defined $\psi(u, v) = (H^{(k)}(u, v) : k \in S)$, i.e., we concatenate the $k$-harmonic distances for all $k \in S$.

**Theorem 6.3.** *Let $[2n] = \{1, 2, \ldots, 2n\}$. For graphs on $n$ vertices, the sparse $[2n]$-harmonic WL test is equally strong as the sparse $\mathbb{R}$-harmonic WL test.*

Lastly, we show that these $[2n]$ harmonic distances subsume all spectral distance measurements with reference to their respective WL tests. That is, any expressive power that can be gained from the spectral distances is contained within the first $[2n]$ $k$-harmonics.

**Theorem 6.4.** *Let $D_f$ be a spectral distance. The sparse $[2n]$-harmonic WL test is as strong as the sparse $D_f$-harmonic WL test.*

In summary, the results in this section imply that there is an important amount of granular expressivity present in the $k$-harmonic distances.

# 7 RESULTS AND DISCUSSION

To corroborate our results, we test the $k$-harmonic distance on both synthetic and real world data. In order to empirically verify the expressivity of the $k$-harmonic distance, we examine performance on the BREC dataset, a dataset that utilizes contrastive learning in order to test a GNN's ability to distinguish non-isomorphic graphs. Further, we examine performance on the ZINC dataset as a measurement for a regression task, and ogbg-molhiv to measure classification. These experiments also validate our theoretical results seen in Section 6, as we see that different datasets have different optimal values for $k$.

**Architecture** We use the $k$-harmonic distances as edge features. We use the GINE architecture (Hu et al., 2020b) as our MPNN. We provide further experimental settings in the Appendix.

**Learnable $k$** In an attempt to further explore the empirical power of the $k$-harmonic distances, we set $k$ to be a learnable parameter in our MPNN as one of our experiments. That is, $k$ is trained with gradient descent during the learning process. In all experiments, we initialize $k = 1.5$, with further justification given in the Appendix.

## 7.1 BREC

The BREC dataset was introduced by Wang & Zhang (2024) as an attempt to measure the *realized* expressivity of GNNs. That is, the dataset makes use of a contrastive learning approach to test whether or not the GNN is able to learn to map non-isomorphic graphs to different features in latent space, and isomorphic graphs to similar areas. The dataset consists of several different types of graphs including Basic Graphs, Regular Graphs, Extension Graphs, and CFI Graphs which range from WL indistinguishable up to 4-WL indistinguishable.

We conduct the experiment by comparing an MPNN that makes use of effective resistance, biharmonic distance, and higher $k$-harmonic distances to their own theoretical expressivity. We present the most compelling results in Table 1, which shows that the realized expressivity of $k$-harmonic distance augmented MPNNs is in line with what we would expect, that is, stronger than 1-WL but weaker than 3-WL. We defer a full discussion of the results to the Appendix.

Further, the learnable $k$ parameter does not prefer any value of $k$ over another. That is, we suspect any value of $k$ results in a local optimum and the MPNN is able to learn BREC nearly equally as

well with any initialization of $k$. This directly supports our findings in Theorem 6.2 that almost any $k$-harmonic works equally as well at distinguishing graphs.

Table 1: % Accuracy for each family of graph in BREC.

| Accuracy | Resistance | Biharmonic | 4-harmonic | Learnable k = arbitrary |
|---|---|---|---|---|
| **Basic** | 100 | 100 | 100 | 98 |
| **Regular** | **50** | 49 | 46 | **50** |
| **Extension** | 95 | **99** | 94 | **99** |
| **CFI** | 4 | 5 | **6** | 4 |
| **Total** | 52 | **52.5** | 51.5 | 52 |

### 7.2 OGBG-MOLHIV

Though BREC provides a measure of the empirical expressivity of different $k$-harmonic distances, it is still imperative to test the performance on real world datasets. We test on ogbg-molhiv, a popular binary classification task that aims to predict whether a given molecule inhibits HIV virus replication.

Table 2 depicts the main results from the experiment. Interestingly, the biharmonic distance has the best absolute performance on the dataset with two layers of message passing. However, when given only one layer of message passing, the learnable $k$ parameter is able to find a more optimal value, tending towards $k = 2$ but ultimately landing on an average of 1.81 across all tests. This suggests that $k = 2$ may be at least a local optimum for $k$. We provide further discussion in the Appendix.

In the spirit of Theorem 6.3, we also concatenate $n$ different $k$-harmonic values together to create an edge feature $e_{uv} \in \mathbb{R}^n$ in order to see whether or not it gives us more expressive power. This experiment is largely ineffective, and single values of $k$ perform better in most instances.

Lastly, we report that these results are statistically significant ($\alpha = 0.05$ via the paired Wilcoxon test). That is, the biharmonic distance is significantly better than any other static value of $k$ as well as not using a $k$ harmonic.

Table 2: % AUC for ogbg-molhiv. $k = [1, 4]$ refers to appending all $k$-harmonic distances from 1 to 4 together. Results are averaged across 10 seeds.

| $k$ | **1 Layer** | **2 Layers** | **4 Layers** |
|---|---|---|---|
| No $k$-Harmonic | $74.2 \pm 1.4$ | $74.3 \pm 1.6$ | $72.5 \pm 3.5$ |
| $k = 1$ | $73.6 \pm 1.6$ | $75.5 \pm 1.3$ | $71.1 \pm 3.7$ |
| $k = 2$ | $75.7 \pm 2.1$ | $\mathbf{78.2 \pm 1.4}$ | $74.4 \pm 2.8$ |
| $k = 3$ | $74.8 \pm 1.5$ | $74.7 \pm 1.7$ | $74.6 \pm 2.4$ |
| $k = 4$ | $73.7 \pm 0.9$ | $72.6 \pm 2.1$ | $70.6 \pm 5.6$ |
| $k = [1, 4]$ | $73.7 \pm 1.3$ | $73.5 \pm 1.7$ | $73.4 \pm 1.4$ |
| Learnable $k$ | $77.0 \pm 1.2$ | $77.5 \pm 1.1$ | $74.1 \pm 1.4$ |

### 7.3 ZINC

The form of the ZINC dataset that we utilize (12K) was formalized in Dwivedi et al. (2023) which is a graph regression task that seeks to learn the constrained solubility of a molecule.

Similarly to molhiv, more layers of message passing help the network with the regression task, though only up to the point of 4 layers. Also of note, the effective resistance outperforms any other $k$-harmonic distance on this dataset. Full results are provided in the Appendix.

The learnable $k$ parameter corroborates our findings and settles at around 1.15 across all experiments, indicating that effective resistance is locally optimal.

Further, we compare to other models across multiple architectures that have used the ZINC dataset in Table 3. The resistance and biharmonic transformer come from (Black et al., 2024b), Graphormer (Ying et al., 2021), GraphGPS (Rampasek et al., 2022), GCN-PE, GAT, 3-WL GNN (Dwivedi et al., 2023). Though our MPNN model is outclassed in raw performance by transformer models that make use of PEs, we argue the benefit of computational efficiency when the power of $k$-harmonic distances are added to low-cost networks — retaining most of the performance with 1/5th of the parameters incurring 1/10th of the runtime (on equivalent hardware) — suggesting that the power lies within the use of the $k$-harmonic distance, rather than the specific choice of network architecture.

Table 3: Test MAE for ZINC compared against number of parameters. The parameter to performance ratio is calculated as $(1/ \text{ test MAE}) \times$ # parameters (in millions), where higher is better.

| | Resistance MPNN | Biharmonic MPNN | Resistance Transformer | Biharmonic Transformer | Graphormer | GraphGPS | GCN-PE | GAT | 3WLGNN |
|---|---|---|---|---|---|---|---|---|---|
| Test MAE | 0.127 | 0.157 | 0.106 | 0.132 | 0.122 | **0.071** | 0.214 | 0.384 | 0.256 |
| # Parameters | **95,601** | **95,601** | 573,922 | 573,922 | 489,321 | 423,717 | 505,011 | 531,345 | 103,098 |
| Performance to Parameter Ratio | **82.36** | 66.63 | 16.44 | 13.20 | 16.75 | 33.24 | 9.25 | 4.90 | 37.89 |

## 8 LIMITATIONS

Though we provide interesting experimental results showing there are better values of $k$ for specific datasets, it is an open question as to why these values of $k$ were better in these cases. That is, though ZINC and ogbg-molhiv are both chemical datasets consisting of graphs that are roughly the same size, our experiments suggest that effective resistance performs better on ZINC while the biharmonic distance performs better on ogbg-molhiv.

Further, though the theoretical expressivity of arbitrary $k$ is clearly present from our BREC results, on real datasets we struggle to find an optimal $k$ that is not 1 or 2 - which is compounded by the inability to have an intuitive interpretation of any $k$-harmonic of $k > 2$. It remains an open question how to interpret values of $k > 2$, and whether or not these $k$-harmonic distanecs have any additional power beyond that of $k = 1, 2$. This is in contrast to the results in (Black et al., 2024a), where values of $k >> 2$ gave the best results for several clustering experiments.

Theorem 6.1 shows that the sparse biharmonic WL can distinguish certain graphs much more quickly than sparse resistance WL. However, sparse resistance WL is able to distinguish these graphs given enough iterations. It is an open question whether there are pairs of graphs that sparse $k$-harmonic WL can distinguish for some value of $k$ but not for another value of $k'$ in any number of iterations.

## 9 CONCLUSION

In this paper, we proved several theoretical properties of the $k$-harmonic distances. We theorized the practical use of these $k$-harmonic distances on specific families of graphs, as well as limitations on the expressive power to be gained with an increasing number of $k$-harmonic distances. To substantiate these results, we provide several empirical tests both on toy datasets and real-world applications to give insight into how these $k$-harmonics perform. We believe this provides a compelling case for the use of the $k$-harmonic distances as a positional encoding in GNNs.

In future work, we hope to further explore exactly what the $k$-harmonic distance tells us about a graph for $k > 2$. By doing so, we may find consistent insights that allows us to recommend specific $k$-harmonic distance for certain types of real world data/graphs.

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

## A PROOFS FROM SECTION 5

### A.1 PROOF OF THEOREM 5.1

**Theorem 5.1.** *Let $k \geq 1$. The sparse $k$-harmonic WL is strictly stronger than the WL test.*

*Proof.* Let $C_n$ denote the ring graph on $n$ nodes. In particular, let $C_9$ be the ring graph on 9 nodes, and let $C_{3,3} = \cup_{i=1}^3 C_3$ be the graph that is the union of 3 ring graphs on 3 nodes; see Figure 1. Observe that $C_9$ and $C_{3,3}$ are indistinguishable by the WL test as they are both 2-regular graphs.

Now, we will show that $C_9$ and $C_{3,3}$ are distinguishable by one iteration of the sparse $k$-harmonic WL.

First, for any edges $e, e' \in C_9$, $H_{C_9}^k(e) = H_{C_9}^{k'}(e')$; likewise for any two edge in $C_{3,3}$. This is because both graphs are edge-transitive. Second, for any edge $e \in E_{C_{3,3}}$ and any edge $e' \in E_{C_3}$, $H_{C_{3,3}}^k(e) = H_{C_3}^{k'}(e')$. This is because for a graph $G$ with connected components $G_1, \ldots, G_k$,

$$L_G^{k+} = \begin{bmatrix} L_{G_1}^{k+} & 0 & \cdots \\ 0 & L_{G_2}^{k+} & \\ \vdots & & \ddots \end{bmatrix}$$

This follows from the fact that the eigenvectors of a disconnected graph are the eigenvectors of each of its connected components with zero padding to be the correct dimesnionality; see (Spielman, 2025, Lemma 3.1.1).

As $C_9$ is a regular graph and all edges have the same $k$-harmonic distance, then all nodes in $C_9$ have the same sparse $k$-harmonic WL color; likewise for $C_{3,3}$. For both graphs, these colors are $\chi^{(1)}(v) = (1, \{\!\{(1, H^k(e), (1, H^k(e)\}\!\})$, where $H^k(e)$ denotes the unqiue $k$-harmonic distance in the respective graph. Therefore, we only need to show that $H_{C_9}^k(e) \neq H_{C_3}^k(e')$ for $e \in E_{C_9}$ and $e' \in C_3$.

We first derive an exact formula for the $k$-harmonic distances of edges in these graphs.

**Lemma A.1.** *Let $C_{2n+1}$ be the cycle graph on $2n+1$ vertices. Let $k > 0$. Then the $k$-harmonic distance of any edge in $C_{2n+1}$ is $H_{C_{2n+1}}^k(e) = \frac{2}{2n+1} \sum_{t=1}^n \left(2 - 2\cos\left(\frac{2\pi t}{2n+1}\right)\right)^{-(k-1)}$*

*Proof.* The analytical form of the eigenvectors and eigenvalues of cycle graphs are well-established (Spielman, 2025, p. 49). In addition to the all-ones vector with eigenvalue 0 (which is an eigenpair of the Laplacian of all graphs), for all $1 \leq t \leq n$, there are two distinct eigenvalues corresponding $x_t$ and $y_t$ corresponding to the eigenvalue $\lambda_t$. Let the vertices of $C_{2n+1}$ as the integers $\{0, ..., 2n\}$. The eigenvectors are

$$\mathbf{x}_t(i) = \sqrt{\frac{2}{2n+1}} \cdot \cos\left(\frac{2\pi t i}{2n+1}\right), \quad \mathbf{y}_t(i) = \sqrt{\frac{2}{2n+1}} \cdot \sin\left(\frac{2\pi t i}{2n+1}\right)$$

with eigenvalue

$$\lambda_t = 2 - 2\cos\left(\frac{2\pi t}{2n+1}\right).$$

Further, the $k^{\text{th}}$ power of the pseudoinverse of $G$ is

$$(L^+)^k = \sum_{t=1}^n (\lambda_t)^{-k} \left(x_t x_t^T + y_t y_t^T\right)$$

Thus, the $k$-harmonic distance of an edge in $C_{2n+1}$ is

$$H_{C_{2n+1}}^k(i, i+1) = (1_i - 1_{i+1})^T L^{+k}(1_i - 1_{i+1})$$

$$= \sum_{t=1}^{n} \left(2 - 2\cos\left(\frac{2\pi t}{2n+1}\right)\right)^{-k} \frac{2}{2n+1}\left[\cos^2\left(\frac{2\pi ti}{2n+1}\right) + \sin^2\left(\frac{2\pi ti}{2n+1}\right)\right.$$

$$- 2\sin\left(\frac{2\pi ti}{2n+1}\right)\sin\left(\frac{2\pi t(i+1)}{2n+1}\right) - 2\cos\left(\frac{2\pi ti}{2n+1}\right)\cos\left(\frac{2\pi t(i+1)}{2n+1}\right)$$

$$\left. + \sin^2\left(\frac{2\pi t(i+1)}{2n+1}\right) + \cos^2\left(\frac{2\pi t(i+1)}{2n+1}\right)\right]$$

Here $0 \le i < 2n$. The choice of $i$ is arbitrary as all edges in $C_{2n+1}$ have the same $k$-harmonic distance. By applying the common trigonometry identities $\sin^2(x) + \cos^2(x) = 1$ and $\cos(x - y) = \cos(x)\cos(y) + \sin(x)\sin(y)$, we arrive at

$$H_{C_{2n+1}}^k(i, i+1) = \frac{2}{2n+1}\sum_{t=1}^{n}\left(2 - 2\cos\left(\frac{2\pi t}{2n+1}\right)\right)^{-k}\left[2 - 2\cos\left(\frac{2\pi t}{2n+1}\right)\right]$$

$$= \frac{2}{2n+1}\sum_{t=1}^{n}\left(2 - 2\cos\left(\frac{2\pi t}{2n+1}\right)\right)^{-(k-1)} \qquad \square$$

Observe that for the case of the disconnected 3-cycle graph ($n = 1$), the $k$-harmonic of an edge is

$$H_{C_3}^k(e) = \frac{2}{3}\left(2 - 2\cos\left(\frac{2\pi}{3}\right)\right)^{-(k-1)}$$

and for the case of the 9-cycle graph ($n = 4$) we have

$$H_{C_9}^k(e) = \frac{2}{9}\left(2 - 2\cos\left(\frac{2\pi}{9}\right)\right)^{-(k-1)} + \frac{2}{9}\left(2 - 2\cos\left(\frac{4\pi}{9}\right)\right)^{-(k-1)}$$

$$+ \frac{2}{9}\left(2 - 2\cos\left(\frac{2\pi}{3}\right)\right)^{-(k-1)} + \frac{2}{9}\left(2 - 2\cos\left(\frac{8\pi}{9}\right)\right)^{-k-1}$$

It is easy to see that

$$H_{C_9}^k(e) = \frac{1}{3}H_{C_3}^k(e) + \frac{2}{9}\left[\left(2 - 2\cos\left(\frac{2\pi}{9}\right)\right)^{-k-1} + \left(2 - 2\cos\left(\frac{4\pi}{9}\right)\right)^{-k-1} + \left(2 - 2\cos\left(\frac{8\pi}{9}\right)\right)^{-k-1}\right]$$

or more simply

$$H_{C_9}^k(e) = \frac{1}{3}H_{C_3}^k(e) + f(k)$$

so we want to show that $f(k) > \frac{2}{3}H_{C_3}^k$. Because $\cos(\pi x)$ is strictly decreasing on the interval $x \in [0, 1]$ this implies that

$$0 < 2 - 2\cos\left(\frac{2\pi}{9}\right) < 2 - 2\cos\left(\frac{4\pi}{9}\right) < 2 - 2\cos\left(\frac{2\pi}{3}\right) < 2 - 2\cos\left(\frac{8\pi}{9}\right)$$

Accordingly, for $k \ge 1$,

$$\frac{2}{9}\left(2 - 2\cos\left(\frac{2\pi}{9}\right)\right)^{-(k-1)} \ge \frac{2}{9}\left(2 - 2\cos\left(\frac{4\pi}{9}\right)\right)^{-(k-1)}$$

$$\ge \frac{2}{9}\left(2 - 2\cos\left(\frac{2\pi}{3}\right)\right)^{-(k-1)} = \frac{1}{3}H_{C_3}^k(e)$$

$$\ge \frac{2}{9}\left(2 - 2\cos\left(\frac{8\pi}{9}\right)\right)^{-(k-1)} > 0$$

The first two terms are larger than $\frac{1}{3}H_{C_3}^k(e)$, with the last term being strictly positive regardless. This implies that

$$f(k) > \frac{2}{3}H_{C_3}^k$$

or that $H_{C_9}^k(e)$ and $H_{C_3}^k(e)$ are never equal for any value of $k \geq 1$. This implies that any value of $k$ used for the sparse $k$-harmonic test will be able to successfully distinguish any two edges in $C_{3,3}$ and $C_9$. Therefore, a single iteration of sparse $k$-harmonic WL will result in different multisets for $G$ and $H$ as the edge features will be aggregated to their incident nodes. Thus, as we have found a pair of graphs that sparse $k$-harmonic WL can distinguish that 1-WL cannot, sparse $k$-harmonic WL > WL. $\qquad\square$

**Corollary A.1.** *Sparse $k$-harmonic WL can distinguish any two odd cycle ring graphs of the form $C_n$ and $\frac{n}{m}$ copies of $C_m$ where $m|n$.*

*Proof.* Observe that the logic in the previous proof follows similarly when $n$ is varied up to

$$\sum_{t=1}^{n} \frac{2}{n}\left(2 - 2\cos\left(\frac{2\pi t}{2n+1}\right)\right)^{-k-1}$$

for the rest of the proof to hold, we need to deduce that $H_{C_m}^k(e) \subset H_{C_n}^k(e)$. For this to be true for two summations of the form $\cos(2\pi t/2n+1)$ it must be the case that $m|n$. While it is true that the two summations will share terms when $m$ and $n$ are not coprime, for $H_{C_m}^k(e) \subset H_{C_n}^k(e)$ it must be that $m|n$.

From here, the rest of the proof remains true. $\qquad\square$

A.2 PROOF OF THEOREM 5.2

We use the following lemma about the number of roots of an exponential function. A stronger variant of it is proved in (Jameson, 2006, Theorem 3.1).

**Lemma A.2.** *Let $f(x) = \sum_{i=1}^{t} a_i b_i^x$, with nonzero $a_i$s and positive $b_i$s. Then, $f(x) = 0$ for at most $t$ values of $x$.*

**Theorem 5.2.** *The 3-WL test is strictly stronger than the sparse $k$-harmonic WL test for all $k \in \mathbb{R}$.*

*Proof.* We rely on the important results from (Zhang et al., 2024), which proves that 3-WL upper-bounds an isomorphism test called the eigenspace projection-WL (EP-WL). We will thus show that sparse $k$-harmonic WL is upper bounded by EP-WL. EP-WL is defined as:

$$\chi_{\mathcal{P}}^{(t+1)}(v) = \left(\chi_{\mathcal{P}}^{(t)}(v), \{\!\!\{\chi_{\mathcal{P}}^{(t)}(u), \mathcal{P}(u,v) : u \in V\}\!\!\}\right) \tag{5}$$

where $\mathcal{P}^L(u,v)$ is the eigenspace projection invariant associated with graph laplacian $L$. Specifically, the Laplacian can be defined as

$$L = \sum_{i \in m} \lambda_i P_i \tag{6}$$

where $\lambda_i$ are the distinct eigenvalues and $P_i$ are the projection matrices for $m$ unique eigenvalues $\lambda_i$. The eigenspace projection invariant is the multiset

$$\mathcal{P}(u,v) = \{\!\!\{(\lambda_1, P_1(u,v)), \ldots, (\lambda_m, P_m(u,v))\}\!\!\}.$$

The outline for the rest of the proof is as follows. We aim to upper bound sparse $k$-harmonic WL by EP-WL. In order to prove our upper bound, we need to prove that both 1.) EP-WL can determine the $k$-harmonic distance of a pair of nodes and 2.) EP-WL can successfully recover which pairs of

nodes are connected by an edge. Part 1) is implied by Lemmas A.3 and A.5 (and proved in the proof of Lemma A.6) and part 2) is Corollary A.2.

We begin with a few observations about EP-WL.

**Lemma A.3.** *Let $G$ and $H$ be graphs. Let $u, v \in V_G$ and $x, y \in V_H$. Then $\mathcal{P}(u, v) = \mathcal{P}(x, y)$ if and only if $L_G^k(u, v) = L_H^k(x, y)$ for all $k \in \mathbb{R}$*

*Proof of Lemma A.3.* If $\mathcal{P}(v, v) = \mathcal{P}(u, w)$, then for any $k$,

$$L_G^k(u, v) = \sum_{i=1}^m \lambda_{G,i}^k P_{i,G}(u, v) = \sum_{i=1}^m \lambda_{H,i}^k P_{i,H}(x, y) = L_H^k(x, y).$$

Now assume $\mathcal{P}(v, v) \neq \mathcal{P}(u, w)$. Consider the polynomial

$$L_G^k(u, v) - L_H^k(x, y) = \sum_{i=1}^m \lambda_{G,i}^k P_{G,i}(u, v) - \sum_{i=1}^m \lambda_{H,i}^k P_{H,i}(x, y)$$

As $\mathcal{P}(v, v) \neq \mathcal{P}(u, w)$, then there is some $i$ such that $\lambda_{i,G} \neq \lambda_{i,H}$ or $P_{G,i}(u, v) \neq P_{H,i}(x, y)$. In either case, this polynomial is not the zero polynomial. Thus, by Lemma A.2, there must be some $k$ such that $L_G^k(u, v) - L_H^k(x, y) \neq 0$, and so $L_G^k(u, v) \neq L_H^k(x, y)$. $\square$

**Corollary A.2.** *Let $G$ and $H$ be graphs. Let $u, v \in V_G$ and $x, y \in V_H$. If $\mathcal{P}(u, v) = \mathcal{P}(x, y)$, then $(u, v) \in E_G$ if and only if $(x, y) \in E_H$*

*Proof of Corollary A.2.* $L_G(u, v) < 0$ if and only if $(u, v) \in E_G$, so this follows from Lemma A.3. $\square$

In what follows, an ***isolated vertex*** is a vertex with 0 neighbors.

**Corollary A.3.** *Let $G$ and $H$ be graph. Let $v \in V_G$ and $u, w \in V_H$. If $v$ is not an isolated vertex, then $\mathcal{P}(v, v) = \mathcal{P}(u, w)$ only if $u = w$.*

*Proof of Corollary A.3.* $L_H(u, w) > 0$ only if $u = w$ and $u$ is not an isolated vertex, so this follows from Lemma A.3. $\square$

**Lemma A.4.** *Let $G$ and $H$ be graph. Let $v \in V_G$ and $x \in V_H$. If $\chi_{\mathcal{P}}^{(1)}(v) = \chi_{\mathcal{P}}^{(1)}(x)$, then either both $v$ and $x$ are isolated vertices or neither $v$ and $x$ are isolated vertices.*

*Proof of Lemma A.4.* If $v$ is an isolated vertex, for any $u \in V$, $L_G(u, v) = 0$. Therefore, as $\{\!\{\mathcal{P}(u, v) : v \in V_G\}\!\} = \{\!\{\mathcal{P}(x, y) : y \in V_H\}\!\}$, by Lemma A.3, it must also be the case that $L_H(x, y) = 0$ for all $y \in V_H$. $\square$

**Lemma A.5.** *Let $G$ and $H$ be graph. Let $v \in V_G$ and $x \in V_H$. If neither $v$ and $x$ are isolated vertices and $\chi_{\mathcal{P}}^{(1)}(v) = \chi_{\mathcal{P}}^{(1)}(x)$, then $L^k(v, v) = L^k(x, x)$ for all $k \in \mathbb{R}$.*

*Proof of Lemma A.5.* If $\chi_{\mathcal{P}}^{(1)}(u) = \chi_{\mathcal{P}}^{(1)}(v)$, then this implies that $\{\!\{\mathcal{P}(u, v) : v \in V_G\}\!\} = \{\!\{\mathcal{P}(x, y) : v \in V_G\}\!\}$. As $v$ and $x$ are not isolated, then by Corollary A.3, $\mathcal{P}(v, v) = \mathcal{P}(x, x)$. Thus, Lemma A.3 implies the lemma. $\square$

Recall that the $k$-harmonic distance is

$$H_k(s, t) = \sqrt{L^{+k}(s, s) + L^{+k}(t, t) - 2L^{+k}(s, t)}$$

Let $\chi_k^{(t)}(v)$ denote the sparse $k$-harmonic WL color.

**Lemma A.6.** *Let $G$ and $H$ be graphs. Let $v \in V_G$ and $x \in V_H$. For all $t \geq 0$, if $\chi_{\mathcal{P}}^{(t+1)}(v) = \chi_{\mathcal{P}}^{(t+1)}(x)$, then $\chi_k^{(t)}(v) = \chi_k^{(t)}(x)$.*

*Proof of Lemma A.6.* We prove this by induction on $t$. For $t = 0$, this is trivial as all vertices have the same sparse $k$-harmonic WL color.

Now assume this is true for some $t - 1$. We will prove it is the case for $t$.

If $\chi_{\mathcal{P}}^{(t+1)}(v) = \chi_{\mathcal{P}}^{(t+1)}(x)$, then by Lemma A.4, there are two cases: both $v$ and $x$ are isolated vertices or neither are.

If $v$ and $x$ are isolated vertices, then $\chi_k^{(t)}(v) = \chi_k^{(t)}(x)$ as all isolated vertices have the sparse $k$-harmonic WL color.

If $v$ and $x$ are not isolated vertices, then

$$\left( \chi_{\mathcal{P}}^{(t)}(v), \{\!\{ \chi(u), \mathcal{P}(u,v) : u \in V_G \}\!\} \right) = \left( \chi_{\mathcal{P}}^{(t)}(x), \{\!\{ \chi(y), \mathcal{P}(x,y) : y \in V_H \}\!\} \right).$$

By the induction hypothesis, the first part of the tuple implies that $\chi_k^{(t-1)}(v) = \chi_k^{(t-1)}(x)$.

Next, observe that by Corollary A.2 that

$$\{\!\{ \chi_{\mathcal{P}}^{(t)}(u), \mathcal{P}(u,v) : u \in V_G \}\!\} = \{\!\{ \chi_{\mathcal{P}}^{(t)}(y), \mathcal{P}(x,y) : (x,y) \in V_H \}\!\}$$

$$\Rightarrow \{\!\{ \chi_{\mathcal{P}}^{(t)}(u), \mathcal{P}(u,v) : (u,v) \in E_G \}\!\} = \{\!\{ \chi_{\mathcal{P}}^{(t)}(y), \mathcal{P}(x,y) : (x,y) \in E_H \}\!\}$$

Thus, there is a bijection $\sigma : N(v) \to N(X)$ such that $(\chi_{\mathcal{P}}^{(t)}(u), \mathcal{P}(u,v)) = (\chi_{\mathcal{P}}^{(t)}(\sigma(u)), \mathcal{P}(x, \sigma(u)))$ for all $u \in N(v)$. We claim that for each $u \in N(v)$ that $(\chi_{k-1}^{(t)}(u), H^k(u,v)) = (\chi_k^{(t-1)}(\sigma(u)), H^k(x, \sigma(u)))$. As $\chi_{\mathcal{P}}^{(t)}(u) = \chi_{\mathcal{P}}^{(t)}(\sigma(u))$, the inductive hypothesis implies that $\chi_{k-1}^{(t)}(u) = \chi_k^{(t-1)}(\sigma(u)$. To prove that $H^k(u,v) = H^k(x, \sigma(u))$, first observe that because $v$ and $x$ are not isolated vertices and $\chi_{\mathcal{P}}^{(t+1)}(v) = \chi_{\mathcal{P}}^{(t+1)}(x)$, then $L_G^{+k}(v,v) = L_G^{+k}(x,x)$ by Lemma A.5. Likewise, $L_G^{+k}(u,u) = L_G^{+k}(\sigma(u), \sigma(u))$. Finally, as $\mathcal{P}(u,v) = \mathcal{P}(x, \sigma(u))$, then $L^{+k}(u,v) = L^{+k}(x, \sigma(u))$ by Lemma A.3. Therefore, $H^k(u,v) = H^k(x, \sigma(u))$. As we have shown there is a bijection $\sigma : N(v) \to N(X)$ such that $(\chi_{k-1}^{(t)}(u), H^k(u,v)) = (\chi_k^{(t-1)}(\sigma(u)), H^k(x, \sigma(u)))$ for all $u \in N(v)$, this concludes our proof that $\chi_k^{(t)}(v) = \chi_k^{(t)}(x)$ $\qquad \square$

We can now use this lemma to prove the theorem. If $G$ and $H$ are 3-WL indistinguishable, they are EP-WL indistinguishable by Zhang et al. (2024). If $G$ and $H$ are EP-WL indistinguishable, this implies that $\{\!\{ \chi_{\mathcal{P}}^{(t)}(v) : v \in V_G \}\!\} = \{\!\{ \chi_{\mathcal{P}}^{(t)}(x) : x \in V_H \}\!\}$ for all $t \geq 0$. Lemma A.6 then implies $\{\!\{ \chi_k^{(t)}(v) : v \in V_G \}\!\} = \{\!\{ \chi_k^{(t)}(x) : x \in V_H \}\!\}$, so $G$ and $H$ are sparse $k$-harmonic WL indistinguishable. $\qquad \square$

# B  PROOFS FROM SECTION SECTION 6

## B.1  PROOF OF THEOREM 6.1

The **$k$-hop neighbor** of radius $k$ around a node $v$ is the graph $(B_k(v), E_k(v))$ with nodes $B_k(v) = \{u \in V : d(v,u) \leq k\}$ and edges $E_k(v) = \{\{u,w\} : d(v,u) \leq k-1, d(v,w) \leq k\}$. Two nodes $u$ and $v$ have **isomorphic $k$-hop neighborhoods** if there is a graph isomorphism $\sigma : B_k(u) \to B_k(v)$ such that $\sigma(u) = v$. While the following lemma is folklore, we will prove a stronger version of this theorem in the coming section (proof of Lemma B.3), so readers interested in a proof of this lemma are encouraged to read that proof.

**Lemma B.1** (Folklore). *Let $G$ and $H$ be graphs, and let $v \in V_G$ and $u \in V_H$. If $v$ and $u$ have isomorphic $k$-hop neighborhoods, then the WL colors $\chi^{(l)}(v) = \chi^{(l)}(x)$ for all $0 \leq l \leq k$.*

**Lemma B.2** ((Black et al., 2024a, Theorem 5.1)). *Let $G = (V, E)$ be a connected graph. Let $(u,v) \in E$ be a cut edge, and let $S, T \subset V$ be the connected components of $G$ after removing the edge $(u,v)$. Then*

$$B(u,v)^2 = \frac{|S||T|}{|V|}.$$

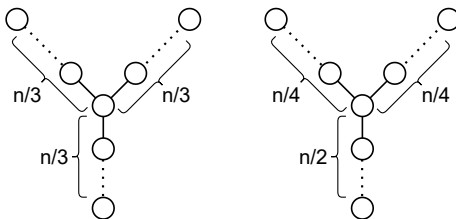

Figure 2: Two non-isomorphic trees that sparse biharmonic WL can distinguish in 1 iteration but sparse resistance WL cannot distinguish in $o(n)$ iterations.

*Proof of theorem 6.1.* Let $n$ be any positive integer that is divisible by $12$. We consider the pair of rooted trees $G$ and $H$ where $G$ is a root connected to three paths of length $n/3$ and $H$ is a root connected to two paths of length $n/4$ and one path of length $n/2$; see Figure 2 for a picture of $G$ and $H$. Observe that these graphs are both trees and both have $n + 1$ vertices. We will show that $G$ and $H$ are indistinguishable by $\lfloor n/8 \rfloor$ iterations of the WL test, but are distinguishable by a single iteration of the sparse biharmonic WL test.

First, we show that these graphs are distinguishable by one iteration of sparse biharmonic WL. First, observe that because $G$ and $H$ are both trees, then all edges in either graph is a cut edge; accordingly, we can use Lemma B.2 to compute the biharmonic distance of all edges in each graph. In particular, consider the edge $e$ connecting the root of $H$ to the path of length $n/2$. The squared biharmonic distance of $e$ is $B(e)^2 = \frac{(n/2)(n/2+1)}{n+1}$; any edge $e'$ in $G$ has biharmonic distance at most $B(e')^2 \le \frac{(n/3)(2n/3+1)}{n+1} < \frac{(n/2)(n/2+1)}{n+1} = B(e)^2$. Therefore, the sparse biharmonic WL color of the root $\chi_B^{(1)}(r_H)$ of $H$ contains an edge with biharmonic distance $B(e) = \sqrt{\frac{(n/2)(n/2+1)}{n+1}}$. As there is no edge in $G$ with biharmonic distance $\sqrt{\frac{(n/2)(n/2+1)}{n+1}}$, there is no node in $G$ with the same sparse biharmonic WL color as $r_H$. Therefore, one iteration of sparse biharmonic WL distinguishes $G$ and $H$.

Next, we need to show that $G$ and $H$ cannot be distinguished in $\lfloor \frac{n}{8} \rfloor$ iterations of the WL test. First, we observe that for any $k < \lfloor \frac{n}{8} \rfloor$, the $k$-hop neighborhoods of the nodes in $G$ and $H$ are of one of three types:

1. Nodes that are distance $r < k$ from a leaf of a tree  The $k$-hop neighborhoods of these nodes are the node connected to a path of length $r$ (the path connecting the node to the leaf) and a path of length $k$.

2. Nodes that are distance $r < k$ from the root  The $k$-hop neighborhoods of these nodes are the node connected to a path of length $k$ and a path of length $r$ connected to two paths of length $k - r$.

3. Nodes that are distance $> k$ from both a leaf and the root  The $k$-hop neighborhood of these nodes are the node connected to two paths of length $k$.

As $k < \lfloor \frac{n}{8} \rfloor$, there are no nodes of that are both distance $< k$ to a leaf and a root, as the distance between any leaf and a root in either tree is $\frac{n}{4}$

For any $0 < r < k$, in both graphs, there are three nodes of distance exactly $r$ to a leaf and distance exactly $r$ to the root. For $r = 0$, there is one node of distance $r$ to the root (the root itself) and three nodes of distance $r = 0$ to the leaves (the leaves themself.) The remaining $n - 6k$ nodes of the graph are at distance $> k$ from both a leaf to a root. As there is a bijection from the nodes of $G$ to the nodes of $H$ such that paired nodes have isomorphic $k$-hop neighborhoods, then by Lemma B.1, we conclude that $G$ and $H$ are indistinguishable by $k$ iterations of the WL test for all $k < \lfloor \frac{n}{8} \rfloor$. $\square$

## B.2 EXAMPLE OF THEOREM 6.1

In Theorem 6.1, we asserted that there are trees that sparse biharmonic WL can distinguish in one iteration. However, this does not generalize to all non-isomorphic trees. We provide one such counterexample in Figure 3 where it would take both sparse biharmonic WL and 1-WL $\Omega(n)$ iterations to distinguish these two trees.

**Theorem B.1.** *There exist pairs of graph $G$ and $H$ with $n$ nodes that cannot be distinguished in $o(n)$ iterations of the sparse biharmonic WL test.*

To prove this, we will use a variant of Lemma B.2 for the sparse $\psi$ WL test. For an edge positional encoding $\psi$ and two graphs $G$ and $H$, we define a **$\psi$-*preserving isomorphism*** as an isomorphism $\sigma : V_G \to V_H$ such that for each edge $(u, v) \in E_G$, $\psi(u, v) = \psi(\sigma(u), \sigma(v))$. In the following lemma, when we say a $\psi$-preserving isomorphism between neighborhoods, we define $\psi$ with respect to the entire graphs $G$ and $H$, and not with respect to the neighborhoods.

**Lemma B.3.** *Let $G$ and $H$ be graphs, and let $v \in V_G$ and $u \in V_H$. Let $\psi$ be an edge positional encoding. If there is a $\psi$-preserving isomorphism between the $k$-hop neighborhoods of $v$ and $x$, then the WL colors $\chi_\psi^{(l)}(v) = \chi_\psi^{(l)}(x)$ for all $0 \leq l \leq k$.*

*Proof.* We will actually prove a stronger result. If there is a $\psi$-preserving isomorphism $\sigma$ between the $k$-hop neighborhoods of $u$ and $v$, then for all $0 \leq l \leq k$ and for all vertices $u$ that are at most $k - l$ hops away from $v$, then $\chi_\psi^{(l)}(v) = \chi_\psi^{(l)}(\sigma(u)))$. We will prove this by induction on $l$. As $u$ is 0 hops from itself, then this implies the theorem.

For the base case of $l = 0$, this is true by the definition of the sparse $\psi$ WL test.

Now assume this is true for some $l \geq 0$; we will prove it is true for $l + 1$. Consider a vertex $u$ that is at distance at most $k - (l + 1)$ from $v$. We claim that $\chi_\psi^{(l+1)}(u) = \chi_\psi^{(l+1)}(\sigma(u))$. The color of $v$ is defined

$$\chi_\psi^{(l+1)}(u) = (\chi_\psi^{(l)}(u), \{\!\!\{(\chi_\psi^{(l)}(u), \psi(u, w)) : (u, w) \in E_G\}\!\!\}).$$

By the inductive hypothesis, we know that $\chi_\psi^{(l)}(u) = \chi_\psi^{(l)}(\sigma(u))$ as $u$ is at most distance $k - (l+1) < k - l$ from $v$.

Moreover, as $\sigma$ is an isomorphism, then the neighbors of $u$ are $\{\sigma(w) : (u, w) \in E_G\} = \{y : (\sigma(u), y) \in E_H\}$. Moreover, any neighbor of $u$ is at most distance $k - l$ from $v$, so $\chi_\psi^{(l)}(w) = \chi_\psi^{(l)}(\sigma(w))$. Finally, as $\sigma$ is $\psi$-preserving, we know that $\psi(u, w) = \psi(\sigma(u), \sigma(w))$. Therefore, we conclude that

$$\begin{aligned}
\chi_\psi^{(l+1)}(u) &= (\chi_\psi^{(l)}(u), \{\!\!\{(\chi_\psi^{(l)}(u), \psi(u, w)) : (u, w) \in E_G\}\!\!\}) \\
&= (\chi_\psi^{(l)}(\sigma(u)), \{\!\!\{(\chi_\psi^{(l)}(\sigma(u)), \psi(\sigma(u), \sigma(w))) : (u, w) \in E_G\}\!\!\}) \\
&= (\chi_\psi^{(l)}(\sigma(u)), \{\!\!\{(\chi_\psi^{(l)}(\sigma(u)), \psi(\sigma(u), y)) : (\sigma(u), y) \in E_H\}\!\!\}) \\
&= \chi_\psi^{(l+1)}(\sigma(u)) \qquad\qquad\qquad\qquad\qquad\qquad\qquad\qquad \square
\end{aligned}$$

*Proof of Theorem B.1.* We will prove this theorem for the two graphs shown in Figure 3. Let $n = 2^k - 1$ for some integer $k$. $G$ and $H$ both consist of a root, two children, and then attached to each child a path of length $n$. At the end of the path, there is either a complete binary tree containing $n$ vertices or a path of length $n$. $G$ and $H$ are not isomorphic as the binary trees in either graphs have different least common ancestors.

Observe that all edges in both $G$ and $H$ are a cut, so by Lemma B.2, the biharmonic distance of *any* edge will equal $|S||T|/|V|$, where $S$ and $T$ are sets of vertices on either side of the cut. Each branch of both $G$ and $H$ has the same number of nodes.

Therefore, it should be easy to see that for any edge in $G$, there is an edge in $H$ must have the same biharmonic distance. These edges are matched in the "obvious" way, i.e., each edge in the tree $T$ is matched to the edge in the same position in the other tree. Moreover, if we match the nodes in $G$ and $H$ in the "obvious" way, then the $k$-hop neighborhoods of these nodes will be biharmonic-preserving

isomorphic for all $k < n$. In other words, for $k < n$, no $k$-hop neighborhood can tell if two trees, two paths, or a tree and a path are in the same branch of the tree. Therefore, by Lemma B.3, $G$ and $H$ cannot be distinguished in $n$ iterations of the sparse biharmonic WL test. □

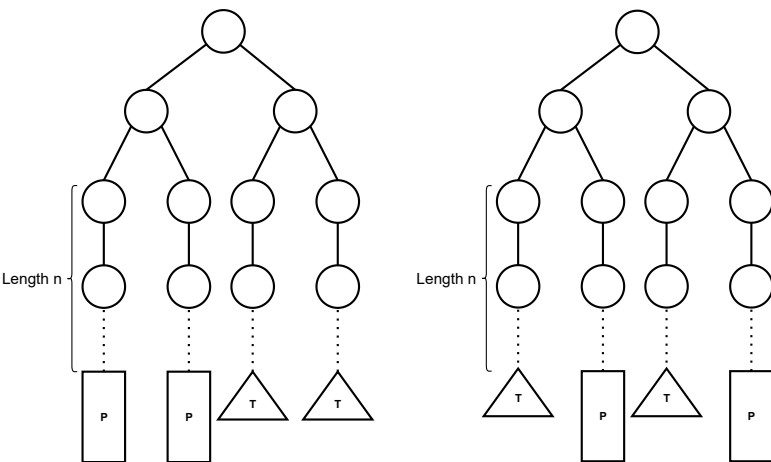

Figure 3: Two non-isomorphic trees $G$ and $H$ with $n$ vertices that sparse biharmonic WL takes $o(n)$ iterations to distinguish. Let $T$ be the complete tree consisting of $n$ nodes and $P$ be a path of $n$ nodes.

### B.3 PROOF OF THEOREM 6.2

**Theorem B.2.** *Let $G$ and $H$ be two graphs such that the Laplacians $L_G$ and $L_H$ have $t_G$ and $t_H$ distinct non-zero eigenvalues respectively. Let $u, v \in V_G$ and $x, y \in V_H$. Then, either*

*(1) the $k$-harmonic distances $H^k(u, v) = H^k(x, y)$ for all $k \in \mathbb{R}$, or*

*(2) the $k$-harmonic distances $H^k(u, v) = H^k(x, y)$ for at most $t_G + t_H$ values of $k \in \mathbb{R}$.*

*Proof.* Suppose that the Laplacian of $G$ has $t_G$ distinct non-zero eigenvalues. Let $L_G$ be the Laplacian of $G$, let $0 < \lambda_1 < \lambda_2 < \cdots < \lambda_t$ be its nonzero distinct Laplacian eigenvalues, let $U_i$ be the matrix with columns that are an orthogonal basis for the eigenspace of $U_i$. Then the Laplacian of $G$ is

$$L_G = \sum_{i=1}^{t} \lambda_i U_i U_i^T.$$

Therefore, the $k$-harmonic distance between two vertices $u$ and $u$ is

$$(1_u - 1_v)^T (L_G^+)^k (1_u - 1_v) = (1_u - 1_v)^T \left( \sum_{i=2}^{n} \lambda_i^{-k} U_i U_i^T \right) (1_u - 1_v)$$

$$= \sum_{i=1}^{t} \lambda_i^{-k} (U_i^T (1_u - 1_v))^T U_i^T (1_u - 1_v)$$

$$= \sum_{i=1}^{t} \lambda_i^{-k} p_i^2(u, v),$$

where $p_i(x, y) = \|U_i^T (1_x - 1_y)\|_2$.

We can similarly write $L_H$ as the decomposition of its eigenvalues and eigenvectors. To distinguish the eigenvectors and eigenvalues of $G$ and $H$, we will denote each with a subscript $G$ and $H$

It follows that if $G$ and $H$ have the same number $t$ of distinct eigenvalues, these eigenvalues are equal ($\lambda_{i,G} = \lambda_{i,H}$ for $1 \leq i \leq t$), and $p_i(u,v) = p_i(x,y)$ for all $1 \leq i \leq t$, then these pairs must have the same $k$-harmonic distance for all values of $k$.

Otherwise, either $G$ and $H$ have a different number of distinct eigenvalues, there is an eigenvalue $\lambda_{G,i} \neq \lambda_{H,i}$, or there exists at least one $2 \leq i \leq t$ for which $p_i(x,y) \neq p_i(u,v)$. Now, consider

$$f(k) = (1_x - 1_y)^T (L_H^+)^k (1_x - 1_y) - (1_u - 1_v)^T (L_G^+)^k (1_u - 1_v) = \sum_{i=1}^{t_G} \lambda_{i,G}^{-k} p_i^2(u,v) - \sum_{i=1}^{t_H} \lambda_{i,H}^{-k} p_i^2(x,y)$$

as a function $k \in \mathbb{R}$, i.e. $f : \mathbb{R} \to \mathbb{R}$ and $k$ is its only variable. Since this is an exponential function that is not identically zero, it has at most $t_G + t_H$ roots by Lemma A.2. The $k$-harmonic distances between $x,y$ and $u,v$ are different for all other values of $k$. $\qquad\square$

**Lemma B.4.** *Let $G$ and $H$ be graphs. Let $\psi$ and $\psi'$ be edge positional encodings such that, for all $u, v \in V_G$ and $x, y \in V_H$, $\psi(u,v) \neq \psi(x,y)$ implies $\psi'(u,v) \neq \psi'(x,y)$. Then if sparse $\psi$ WL distinguishes $G$ and $H$, then sparse $\psi'$ WL distinguishes $G$ and $H$.*

*Proof.* Let $\chi_\psi^{(t)}(v)$ denote the color of node $v$ under the sparse $\psi$ WL test at step $t$. Further, let $v_i \in G$ and $v_j \in H$. As a first step towards proving this lemma, we will show that if $\chi_\psi^{(t)}(v_i) \neq \chi_\psi^{(t)}(v_j)$, then $\chi_{\psi'}^{(t)}(v_i) \neq \chi_{\psi'}^{(t)}(v_j)$, or that if nodes $v_i$ and $v_j$ are not the same color under the sparse $\psi$ WL test, they will not be the same color under the sparse $\psi'$ WL test. We will prove this by induction on $t$.

Base Case: For $t = 0$, this is vacuously true as $\chi_\psi^{(0)}(v_i) = \chi_\psi^{(0)}(v_j) = 1$ for all $v_i \in G$ and $v_j \in H$.

Induction Hypothesis: Suppose this is true for $t - 1 \geq 0$. That is, if $\chi_\psi^{(t-1)}(v_i) \neq \chi_\psi^{(t-1)}(v_j)$, then $\chi_{\psi'}^{(t-1)}(v_i) \neq \chi_{\psi'}^{(t-1)}(v_j)$

We will now prove this is true for $t$. Suppose $\chi_\psi^{(t)}(v_i) \neq \chi_\psi^{(t)}(v_j)$. By definition of the WL test, it is either the case that: the colors were different in the previous iteration of the test: $\chi_\psi^{(t-1)}(v_i) \neq \chi_\psi^{(t-1)}(v_j)$, or the nodes aggregated distinguishing information in step $t$: $\{\!\{\psi(v_i, x), \chi_\psi^{(t-1)}(x) : (v_i, x) \in E_G\}\!\} \neq \{\!\{\psi(v_j, y), \chi_\psi^{(t-1)}(y) : (v_j, y) \in E_H\}\!\}$

- Case 1: $\chi_\psi^{(t-1)}(v_i) \neq \chi_\psi^{(t-1)}(v_j)$ By the induction hypothesis, $\chi_{\psi'}^{(t-1)}(v_i) \neq \chi_{\psi'}^{(t-1)}(v_j)$. This implies that $\chi_{\psi'}^{(t)}(v_i) \neq \chi_{\psi'}^{(t)}(v_j)$

- Case 2: $\{\!\{\psi(v_i, x), \chi_\psi^{(t-1)}(x) : (v_i, x) \in E_G\}\!\} \neq \{\!\{\psi(v_j, y), \chi_\psi^{(t-1)}(y) : (v_j, y) \in E_H\}\!\}$
  WLOG suppose that $v_i$ and $v_j$ have the same number of neighbors, as their multisets will be vacuously different if they don't. Given that they have the same number of neighbors but have different multisets of colors, we can conclude that for any bijection $\sigma : N(v_i) \to N(v_j)$, there is a vertex $u \in N(v_i)$ such that $(\psi(v_i, u), \chi_\psi^{(t-1)}(u)) \neq (\psi(v_j, \sigma(u)), \chi_\psi^{(t-1)}(\sigma(u)))$.

  If $\chi_\psi^{(t-1)}(u) \neq \chi_\psi^{(t-1)}(\sigma(u))$ then the induction hypothesis holds and $\chi_{\psi'}^{(t-1)}(u) \neq \chi_{\psi'}^{(t-1)}(\sigma(u))$. If $\psi(v_i, u) \neq \psi(v_j, \sigma(u))$, then we invoke our assumption to say $\psi'(v_i, u) \neq \psi'(v_j, u)$ and the statement holds.

Thus, in both cases, $\chi_{\psi'}^{(t)}(v_i) \neq \chi_{\psi'}^{(t)}(v_j)$.

To finish the proof, we show that $G$ and $H$ are distinguishable by the sparse $\psi'$ WL test. Given that $G$ and $H$ are distinguishable by sparse $\psi$ WL, there is some $t > 0$ such that $\{\!\{\chi_\psi^{(t)}(v_i) : v_i \in V_G\}\!\} \neq \{\!\{\chi_\psi^{(t)}(v_j) : v_j \in V_H\}\!\}$. So, for any bijection $\sigma : V_G \to V_H$, there is a vertex $v \in V_G$

such that $\chi_\psi^{(t)}(v) \neq \chi_\psi(\sigma(v))$. From the above, this implies that $\chi_{\psi'}^{(t)}(v) \neq \chi_{\psi'}(\sigma(v))$. Therefore, $\{\!\{\chi_{\psi'}^{(t)}(v_i) : v_i \in V_G\}\!\} \neq \{\!\{\chi_{\psi'}^{(t)}(v_j) : v_j \in V_H\}\!\}$, so sparse $\psi'$ WL also distinguishes $G$ and $H$. $\qquad\square$

**Note** In the following theorems, some of the constants in the theorem statements are slightly different than in the body of the paper as we had to revise these proofs in finishing the appendix. However, the statements of these theorems are the same beyond these small changes.

**Theorem 6.2.** *Let $G$ and $H$ be graphs with $n$ vertices that are distinguishable by sparse $k$-harmonic WL for some $k$. Then for all but $O(n^5)$ values of $k' \in \mathbb{R}^+$, $G$ and $H$ are distinguishable by the sparse $k'$-harmonic WL test.*

*Proof.* Let $G$ and $H$ be graphs that are distinguishable by $k$-harmonic WL. For all pairs of nodes $u, v \in V_G$ and $x, y \in v_H$, let

$$K(u,v,x,y) = \begin{cases} \emptyset & \text{if } H^k(u,v) = H^k(x,y) \text{ for all } k \in \mathbb{R} \\ \{k \in \mathbb{R} : H^k(u,v) = H^k(x,y)\} & \text{otherwise} \end{cases}$$

Now let $K = \cup_{u,v \in V_G, \, x,y \in V_H} K(u,v,x,y)$ and let $k' \in \mathbb{R} \setminus K$. By construction, for any pairs $u, v \in V_G$ and $x, y \in V_H$, if $H^k(u,v) \neq H^k(x,y)$, then $H^{k'}(u,v) \neq H^{k'}(x,y)$. Therefore, the $k'$-harmonic distance satisfies the conditions of Lemma B.4, so $G$ and $H$ are distinguishable by the sparse $k'$-harmonic WL test. Moreover, $G$ and $H$ each have at most $n-1$ distinct eigenvalues, so the size of $K$ is $O(n^5)$. $\qquad\square$

**Theorem 6.3.** *Let $[2n] = \{1, 2, \ldots, 2n\}$. For graphs on $n$ vertices, the sparse $[2n]$-harmonic WL test is equally strong as the sparse $\mathbb{R}$-harmonic WL test.*

*Proof.* Let $[n] = \{1, 2, \ldots, n\}$. For graphs on $n$ vertices, the sparse $[2n]$-harmonic WL test is equally as strong as the sparse $\mathbb{R}$-harmonic WL test.

Recall that for two WL tests to be equally as strong as one another, if two graphs, $G$ and $H$ are indistinguishable by $x$ then they are indistinguishable by $y$, and vise versa. Another way to say this is if $G$ and $H$ are distinguishable by $x$ then they are distinguishable by $y$, and vise versa.

We invoke the result of Theorem B.2 which implies that either the $H_G^k(u,v) = H_H^k(x,y)$ for all $k \in \mathbb{R}$, or that $H_G^k(u,v) = H_H^k(x,y)$ for at most $2n-2$ values of $k$. As we are working with sparse $[2n]$ WL, if $H_G^k(u,v) = H_H^k(x,y)$ for $1 \leq i \leq 2n$, it must be the case that $H_G^k(u,v) = H_H^k(x,y)$ for all $k \in \mathbb{R}$.

This means that we can invoke Lemma B.4 and say that if sparse $[2n]$-harmonic WL distinguishes $G$ and $H$, then sparse $\mathbb{R}$-harmonic WL can distinguish $G$ and $H$. This implies that sparse $\mathbb{R}$ WL is at least as strong as sparse $[n]$-harmonic WL.

The other direction follows easily. That is, if $H_G^{\mathbb{R}}(u,v) = H_G^{\mathbb{R}}(x,y)$, then vacuously $H_G^k(u,v) = H_H^k(x,y)$ for $1 \leq i \leq 2n$, so we can apply Lemma B.4 in the other direction. $\qquad\square$

**Theorem 6.4.** *Let $D_f$ be a spectral distance. The sparse $[2n]$-harmonic WL test is as strong as the sparse $D_f$-harmonic WL test.*

*Proof.* Let $D_f$ be a spectral distance. We will prove the sparse $[2n]$-harmonic WL test is as strong as the sparse $D_f$ WL test. By Lemma B.4, it is only the case that $H_G^k(u,v) = H_H^k(x,y)$ for $1 \leq k \leq 2n$ if $H_G^k(u,v) = H_H^k(x,y)$ for all $k \in \mathbb{R}$. However, by the proof of Lemma B.4, it is only the case that $H_G^k(u,v) = H_H^k(x,y)$ for all $k \in \mathbb{R}$ if $G$ and $H$ have the same number $t$ of distinct eigenvalues, $\lambda_{i,G} = \lambda_{i,H}$ for $1 \leq i \leq t$, and $p_i(u,v) = p_i(x,y)$ for all $1 \leq i \leq t$. However, if all three of these conditions are true, then for any spectral distance $D_f$, $D_f(u,v) = \sum_{i=1}^t f(\lambda_{i,G})p_i(u,v) = \sum_{i=1}^t f(\lambda_{i,h})p_i(x,y) = D_f(x,y)$. Therefore, by Lemma B.4, the sparse $[2n]$-harmonic WL is as strong as the sparse $D_f$ WL test for any function $f$. $\qquad\square$

## C EXPERIMENTS

**Learnable $k$-Harmonic Distance**  In all reported experiments we initialize $k = 1.5$, chosen as a midpoint between the commonly observed best results for each dataset ($k = 1, 2$). This choice avoids predisposing the model towards either value.

While we experimented with alternative initializations ($k = 0$, $k = 3$, $k \sim \mathcal{U}(0, 1)$), they consistently converge to similar values but yielded inferior performance, likely due to slower convergence and difficulty fitting training data early on.

### C.1 BREC

The BREC dataset includes several families of graphs ranging from 1-WL indistinguishable, to 4-WL indistinguishable. We provide a quick overview of the dataset and justification for why the results we received are consistent with our theoretical results.

**Basic:**  Consists of 60 pairs of 1-WL indistinguishable graphs.

**Regular:**  Consists of 140 pairs of regular graphs, subdivided into different families of regular graph. 50 pairs of simple regular graphs which are 1-WL indistinguishable, 50 pairs of strongly regular graphs which are 3-WL indistinguishable, 20 pairs of 4-vertex condition graphs which are *at least* 3-WL indistinguishable, and 20 pairs of distance regular graphs which are *at least* 3-WL indistinguishable.

**Extension:**  Consists of 100 pairs of graphs that sit between 1-WL indistinguishable and 3-WL distinguishable. These graphs were generated outside of the context of the WL hierarchy with methods such as substructure counting, node marking, and $n$-hop subgraphs. The authors claim that these graphs are meant to provide more granularity to the space between 1-WL and 3-WL.

**CFI:**  Consists of 100 pairs of graphs generated by the intentionally difficult Cai, Furer, and Immerman method. 60 pairs of these graphs are 1-WL indistinguishable, 20 pairs are 3-WL indistinguishable, and a further 20 pairs are 4-WL indistinguishable.

Given that we have previously proven that sparse $k$-harmonic WL is strictly more expressive than 1-WL, but upper bound by 3-WL, we would expect any MPNN equipped with $k$-harmonic distance to distinguish some amount of the graphs in BREC that are 1-WL indistinguishable, but none of the graphs that are 3-WL indistinguishable and higher. Of note, it is well known that 1-WL = 2-WL (Huang & Villar, 2021), so the tightest bound that can be achieved in the WL framework is between 1-WL and 3-WL.

Concretely, sparse $k$-harmonic WL is able to distinguish all 1-WL indistinguishable graphs barring the exceptionally difficult CFI graphs. However, this is not without precedent. Black et al. (2024b) execute a similar experiment with transformer-based models that make use of effective resistance and experience similar difficulties learning the CFI graphs while being able to learn all other 1-WL indistinguishable graphs.

Further, Table 4 shows how both effective resistance and biharmonic distance respond to more layers of message passing. That is, layers of message passing largely does not have an effect on results. It is worth noting that in all experiments, the effective resistance is required to be normalized. That is, without input normalization, effective resistance scores an average $41.75\%$ accuracy on BREC, as opposed to the $52\%$ pictured. This is the only $k$-harmonic that improves with input normalization, and it is unclear why this is the case. Perhaps most interestingly, the layers of message passing do not have a significant effect on the realized expressivity of any $k$'s ability to distinguish graphs.

### C.2 OGBG-MOLHIV

Notably, molhiv also provides edge features indicating the type of bond present between two nodes (atoms), and node features that denote atom type, chirality, among other things. In our early experiments, we see that bond types notably help the $k$-harmonic distances perform in all cases ($\alpha = 0.05$) and thus include them in all experiments. Specifically, the bond types are passed through a learnable

Table 4: % Accuracy for each family of graph in BREC broken down by number of message passing layers for both Effective Resistance and Biharmonic Distance

| | Resistance | | | | Biharmonic | | | |
|---|---|---|---|---|---|---|---|---|
| **Layers** | 1 | 2 | 3 | 4 | 1 | 2 | 3 | 4 |
| **Basic** | 100 | 96.6 | 100 | 100 | 100 | 100 | 96.6 | 96.6 |
| **Regular** | 50 | 50 | 49 | 47 | 46 | 47 | 49 | 49 |
| **Extension** | 95 | 97 | 94 | 95 | 95 | 99 | 93 | 94 |
| **CFI** | 3 | 3 | 4 | 4 | 4 | 6 | 6 | 5 |
| **Total** | 52 | 52 | 51.75 | 51.5 | 51.25 | 52.5 | 51.5 | 51.5 |

linear layer and then summed with the $k$-harmonic distances that have also been passed through a separate learnable linear layer.

Perhaps most importantly, we see that only $k = 1$ and $k = 2$ give statistically significantly better results than the control experiment ($\alpha = 0.05$ via the paired Wilcoxon test). Though we ran several tests experimenting with input normalization (mean, min/max, log), very few had any positive effect on the results. Therefore, it is likely that higher values of $k$ introduce too much numerical instability, or simply lose too much structural information on this dataset. This is further corroborated by the learnable $k$ parameter tending towards 2 with little exploration or variance beyond $k = 2.1$, suggesting that $k$ values beyond 2 are simply suboptimal.

## C.3 ZINC

We present the full results from the ZINC experiments in Table 5, in a similar fashion to Table 2. Consistent with the learnable $k$ parameter settling around $1.15$ across all experiments, we see that effective resistance is likely the optimal value of $k$ for this dataset over biharmonic or any other $k$. Further, we verify that these results are statistically significant ($\alpha = 0.05$ via the paired Wilcoxon test)

Table 5: MAE for ZINC. Results are averaged across 10 seeds.

| $k$ | **1 Layer** | **2 Layers** | **4 Layers** |
|---|---|---|---|
| $k = 1$ | $0.244 \pm 0.005$ | $0.144 \pm 0.005$ | $\mathbf{0.127 \pm 0.004}$ |
| $k = 2$ | $0.368 \pm 0.017$ | $0.188 \pm 0.006$ | $0.157 \pm 0.006$ |
| $k = 3$ | $0.401 \pm 0.008$ | $0.319 \pm 0.020$ | $0.495 \pm 0.417$ |
| $k = 4$ | $0.504 \pm 0.063$ | $0.797 \pm 0.493$ | $1.133 \pm 0.434$ |
| learnable $k$ | $0.218 \pm 0.047$ | $0.142 \pm 0.009$ | $0.136 \pm 0.003$ |

Further, we give a brief summary of extraneous experiments. That is, the inclusion of the edge features that are native to the ZINC dataset (bond information) largely have no effect on results, so we do not include them in our experimentation. Further, input normalization has very little effect in most cases, and is statistically insignificant in all cases regardless of the type of normalization used (min/max, logarithmic, or mean/standard deviation).

**Settings, Hyperparameters, and Hardware** The settings and hyperparameters for any given experiment are contained in the configuration files that accompany our code.

All experiments were run on a single NVIDIA V100 GPU with 32GB of VRAM.

**Code** Our code builds off of (Rampasek et al., 2022; Black et al., 2024b; Müller et al., 2024) and use of the code is allowable under the MIT Licensing present

```
https://anonymous.4open.science/r/expressive_power_of_k_harmonic_
distance-9B0C
```

