# OpenReview forum: "The Expressive Power of k-Harmonic Distances for Message Passing Neural Networks"
_ICLR.cc/2026/Conference — Submitted to ICLR 2026_

### Official Review · Reviewer_TrBu · 2025-10-19

**Soundness:** 3
**Presentation:** 3
**Contribution:** 3
**Rating:** 6
**Confidence:** 3

**Summary:**

This paper studies k-harmonic distances as positional encodings for MPNNs to enhance expressiveness. The authors introduce the sparse $\psi$-WL test framework and establish theoretical bounds showing that k-harmonic distances increase MPNN expressivity beyond 1-WL but not beyond 3-WL.  Experiments on BREC, ZINC, and ogbg-molhiv validate basic claims but reveal that $k \in \\{1,2\\}$ appear sufficient in practice, with higher k values causing performance degradation.

**Strengths:**

1. Strong theoretical framework: The sparse $\psi$-WL test provides a clean formalism for analyzing MPNNs with edge features. This extends prior WL-based expressivity analysis in a principled way and could be useful beyond k-harmonic distances.

2. Novel theoretical insights: Theorems 6.2-6.4 are nolve. The result that almost all k values are equivalent and that 2n values suffice to capture all spectral distances provides new understanding of the spectral distance landscape. These contributions advance the theoretical understanding of spectral positional encodings.

3. Rigorous mathematical treatment: The proofs appear technically sound. The connection to eigenspace structure  and the use of polynomial root bounds are elegant. The paper carefully distinguishes between different notions of expressivity.

**Weaknesses:**

1. The paper's main theoretical contribution is about general k values, yet experiments consistently show only $k \in \\{1,2\\}$ work well. Table 5 shows k=3,4 are worse than baseline on molhiv. The authors acknowledge "numerical instability" but provide no solution.

2. Theorem 6.1 with the main result showing different k values have different power, only applies to a specific pair of tree graphs. The counterexample in Theorem B.1 immediately shows this doesn't generalize.

**Questions:**

1. What about MPNN with just k=1 and k=2 concatenated (not 1 to 4)? Does this work better than either alone?

---

> ### Author Response · Authors · 2025-11-21
>
> We greatly appreciate your feedback and assessment of our manuscript.
>
> -To expand on “numerical instability”, as k grows larger, the values for the k-harmonic distances on edges become vanishingly small, or explode in magnitude (as we are taking the pseudoinverse to higher powers of itself). This causes the relative scaling of the distances to become more difficult to deal with in practice, and also causes the node features to dominate the message passing schema (as evidenced by k=3,4 performing well on BREC, which has no node features). So, while the theoretical expressivity of the k-harmonics holds, the challenges of working with large k empirically generally yield inferior results.
> While hypothetically this should be solvable with standard normalization methods (such as min/max scaling, log scaling, mean scaling), the several that we tried resulted in only marginal performance gains over the unnormalized inputs, with none reaching the performance of k=1,2.
> Therefore, it is unclear to us how much of the performance of the 3- and 4-harmonics can be attributed to their theoretical expressivity versus other factors in training like numerical issues
>
> -While this is true, these two results taken together still imply that biharmonic-WL is faster than resistance-WL on the family of all trees. We admit that this is not readily apparent, and will update the manuscript to be explicit.
> Further, the message behind Section 6 was that not all k-harmonics are trivially equal to one another, despite our results in Section 5 showing that they occupy the same class of expressivity between 1-WL and 3-WL. That is, there is an expressive granularity present that makes it worthwhile to explore what inductive biases might exist in graph data that makes some k preferable to others (such as our tree example).
>
> -In our initial round of experiments that we ran on MOLHIV, the answer to this question was no. However, upon reading your comment we ran further tests experimenting with the concatenation of k=1,2 and found that it indeed yields better results on ZINC. That is, when the k=1,2 are concatenated together, we see an average Test MAE of 0.106 as opposed to the strongest individual k (1) having an average Test MAE of 0.127. Thus, given these results, we would like to run a further hyperparameter sweep on MOLHIV, and we hope to update you in the following week. We thank you for your suggestion.

---

> > ### Author Response · Authors · 2025-12-01
> >
> > Update:
> >
> > We find that the concatenation of k=1,2 does not yield better results on MOLHIV. As a reminder, k=1 gives us 75.5% AUC, and the superior k=2 gives us 78.2%. The resulting concatenation of k=1,2 gives us 75.6% average across 5 seeds. Thus, it appears as though this concatenation yields positive results for ZINC, but the same does not hold to MOLHIV. Further, k=1,2 does perform better than k = [1,4]. This paints an interesting picture for the concatenation of different k-harmonics (and more generally, the concatenation of spectral encodings), but we leave this to future work.

---

### Official Review · Reviewer_BRcr · 2025-10-22

**Soundness:** 3
**Presentation:** 2
**Contribution:** 2
**Rating:** 4
**Confidence:** 3

**Summary:**

This paper explores theoretical properties of positional encodings in GNNs based on spectral quantities. Specifically, the paper focuses on k-harmonic distances, which generalize effective resistance and biharmonic resistance. The paper studies the separation power of networks enhanced with such positional encodings in terms of WL tests.

**Strengths:**

1. Since positional encoding often significantly improves GNN performance, analyzing the theoretical properties of this method is important.
2. The theoretical analysis seems to be sound, though I did not read proofs in detail.
3. The writing style is clear.

**Weaknesses:**

1. The experimental results are not convincing and do not seem to corroborate or support the theoretical findings.

In general, the claims in this paper are a bit confusing. The experiments on BREC and Theorem 6.2 illustrate that all values of k roughly give the same separation power. On the other hand, the paper emphasizes that keeping k a free parameter is important, that different problems prefer different values of k, and that combining different values of k is important. Table 2 also has indecisive results. Table 3 does not even try to set $k>2$.

The author should clarify better what their message is in regards to the choice of k.

----

2. The proofs of the hierarchy of strength of WL tests are based on finding a specific pair of graphs that can be distinguished by one test but not by the other. While there is no error in this approach, it is not very insightful. A more insightful investigation of different WL tests would structurally characterize all graphs (or a large set of graphs) that cannot be distinguished by one test but can by the other. Consider the following point of view: suppose you find an example of a  pair of graphs that can be distinguished by test1 but not by test2, and you show that test1 is at least as strong as test2. A prior, it is possible that this is the only example of such a pair that exists. In such a case, up to this single pair, both tests are equivalent. Hence, in my opinion, a better characterization of the capabilities of WL tests would consist of finding large families of graphs that one test can separate and the other cannot. This shortcoming is of course present in many papers about GNN expressivity via graph isomorphism tests.

As an example of a good characterization, 1-WL cannot separate graphs that have the same tree homomorphism numbers. More generally, k-WL tests are characterized by homomorphism numbers of motifs with treewidth at most k. I find such characterizations of isomorphism tests much more revealing.

----

3. Theorem 6.1 is quite a weak result. It proves that there is a pair of graphs that can be distinguished quickly by biharmonic WL but only slowly by resistance WL. However, this does not prove that biharmonic WL is faster than resistance WL. A priori, it is possible that there is another pair of graphs that can be distinguished quickly by resistance WL but only slowly by biharmonic WL. Hence, it is not clear what the significance of Theorem 6.1 is. Note that the experiments do not support  any k=1,2 being better than the other. Hence, why isn't there a corresponding theorem showing that there is also a pair of graphs that can be separated quickly by resistance WL but only slowly by biharmonic WL? What is the message behind the asymmetry in Theorem 6.1?

----

4. Line 073: It is not true that computing the full eigendecomposition of L is less efficient than computing the pseudo inverse of L. In practice, both can be estimated in $O(n^3)$ operations. Note that, for example, each iteration of the QR algorithm for the full eigendecomposition takes $O(n^3)$, and in most practical settings the algorithm converges with a few iterations. If you only want to compute the leading eigenvectors, then the complexity is even faster.

**Questions:**

See Weaknesses

---

> ### Author Response · Authors · 2025-11-21
>
> We very much appreciate the time and effort you have put into reviewing our paper.
>
> -To summarize our message: while the k-harmonics theoretical expressivity are all between 1-WL and 3-WL, real world data has inductive biases that seem to prefer one k over another, highlighting the important difference between theoretical and empirical expressivity. Hence, our results on BREC indeed show that all values of k give the same separation power (on toy data explicitly meant to test the WL hierarchy). However, on real data such as ZINC and MOLHIV, we see that this theoretical expressivity does not tell the entire story - and these datasets seem to prefer specific values of k (k = 1 for ZINC and k = 2 for MOLHIV). The experiment where we made k a learnable parameter was to corroborate these results - in that it simply wasn’t by coincidence or weight initialization. Hence, when training real networks it may be worth it to cheaply compute the k-harmonic distance for a preferred value of k, rather than using the entire eigendecomposition (touched on below).
> Further, while we attempt to combine different values of k in our experiments, we see that it simply leads to poorer performance with more numerical instability. While this should lead to more theoretical expressivity (Theorem 6.3 and 6.4) we see that this does not hold in real neural networks.
> To recap: Table 1 shows the realized theoretical expressivity of all k on the toy dataset BREC - corroborating our results from section 5. Table 2 shows that k = 2 is the superior measurement for MOLHIV, corroborating the idea from section 6 that not all k-harmonics are trivially equally expressive. Table 3 both shows that k = 1 is the superior measurement for ZINC, and that the performance to parameter (and runtime) ratio is preferable for MPNNs over transformer-based models. The results for k > 2 for ZINC are contained in the Appendix, in Table 5. We declined to have this as the result in the main paper.
>
> -We agree that finding larger families of graphs based on homomorphism numbers is an interesting direction for future research. However, while we only prove the existence of one counterexample for Theorem 5.1 and Corollary A.1, our results on BREC are able to experimentally show that this holds beyond the cycle graphs, so it is not the case this result only holds for this narrow class of graphs. The BREC benchmark consists of graphs that are at least 1-WL indistinguishable, and the k-harmonic distances are able distinguish many graphs that are not 3-WL indistinguishable, which are all graphs that Theorem 5.2 does not explicitly rule out. Moreover, because BREC samples from several different graph families, if our k-harmonics failed on any particular family of graphs, we would see this in the benchmark. Lastly, the authors of BREC explicitly argue that their Extension subset provides this finer granularity in the space between 1-WL and 3-WL by including graphs generated independently of WL using methods such as substructure counting, and node marking, all of which the k-harmonics are able to distinguish.
> As you mentioned this is a shortcoming in many papers of this type. But we do want to point out that the results on BREC indicate that the k-harmonic distances can distinguish many types of graphs.
>
> -While not explicitly stated, Theorem 6.1 taken in conjunction with Theorem B.1 implies that biharmonic-WL is faster than resistance-WL on trees. We admit that this is not readily apparent, and will edit the manuscript to make this explicit.
> Further, the message here is simply that there is expressive granularity in the k-harmonic distances - that they are not trivially equivalent to one another, despite being a part of the same class of expressivity between 1-WL and 3-WL. This motivates our experimental section, where we indeed find that there are cases in the real world where k = 1 is significantly preferable to k = 2 and vice versa. We agree that a similar proof for a pair of graphs that resistance-WL can distinguish quickly where biharmonic-WL cannot would be desirable.
>
> -We agree with your statement and we admit that this is an oversight on our part. Though we do not mention it in the manuscript, we do not need to compute the full pseudo-inverse of the Laplacian to compute some k-harmonic. They can be approximated using the Johnson-Lindenstrauss projection of the Laplacian in nearly linear time [1,2]. We will update the manuscript so that this is explicitly clear - we prefer the k-harmonics for this nearly linear preprocessing cost as opposed to the full eigendecomposition being on the order of O(n^3).
>
> [1] Z. Zhang, W. Xu, Y. Yi and Z. Zhang, "Fast Approximation of Coherence for Second-Order Noisy Consensus Networks," in IEEE Transactions on Cybernetics, 2020.
>
> [2] Spielman, Daniel A., and Nikhil Srivastava. "Graph sparsification by effective resistances." Proceedings of the fortieth annual ACM symposium on Theory of computing. 2008.

---

### Official Review · Reviewer_9Eb6 · 2025-10-25

**Soundness:** 3
**Presentation:** 2
**Contribution:** 2
**Rating:** 2
**Confidence:** 3

**Summary:**

This paper investigates $k$-harmonic distances as positional encodings for message passing graph neural networks (MPNNs), which are used as edge features (i.e., relative positional encodings). The theoretical contribution focuses on analyzing the expressivity of $k$-harmonic distances for varying $k$. To this end, the authors introduce a modification of the 1-WL test that incorporates edge features (sec. 4), and show that augmenting MPNNs with $k$-harmonic distance features makes them strictly more expressive than 1-WL but bounded by 3-WL (sec. 5). The main theoretical contributions are in sec. 6, where the authors show that $k=2$ (i.e., sparse biharmonic WL) can distinguish some graphs in much less iterations than when setting $k=1$ (i.e., sparse resistance WL). They also argue (in Thm. 6.2) that if two graphs can be distinguished by one choice of $k$, this holds for most others as well, and taking all $k \in \{1, \dots, 2n\}$ suffices to be just as expressive as taking all possible $k$. These findings are then corroborated empirically: using BREC to gauge expressivity, and evaluating on real-world datasets (ZINC and ogbg-molhiv) to show that the optimal $k$ depends on the dataset. The authors also experiment with learning $k$, which reveals that different datasets favor different $k$ values in practice.

**Strengths:**

- The theoretical insights are clear, and I like the quantitative nature of Thm. 6.1 (i.e., taking the number of WL iterations into account). The practical limit (for including multiple values of $k$) from Thm. 6.3 is also an interesting result.
- The proofs are well-written and easy to follow; generally from my impression, the paper’s mathematical rigor is high.
- The experimental evaluation is relatively diverse. The authors validate theoretical claims on the BREC synthetic benchmark and demonstrate the method on real-world tasks (ZINC and ogbg-molhiv), showing how different choices of $k$ perform on different types of graphs. This blend of synthetic and real data experiments strengthens the paper’s conclusions. Also, introducing a learnable $k$ in the model provides good insight that there is no one optimal choice.

**Weaknesses:**

- The motivation of looking at MPNNs with dense relative positional encodings is not entirely clear to me. If one has to invert the Laplacian and compute the $k$-harmonic distance (i.e. an $n \times n$ *dense* matrix, this preprocessing step would dominate the runtime for large graphs. In this case, one might ask whether using a graph transformer with global message passing and the same relative PEs would yield a more expressive architecture in practice. As such, this seems like a natural baseline that could be discussed more explicitly. I only found such a comparison in Table 3, but I think this should also be compared against in the other experiments. Directly adjacent to this, there is no discussion of computational overhead or scalability of the method.
- The empirical improvements from $k$-harmonic features are modest and dataset-dependent. On BREC, results for different $k$ values are nearly identical (even with learnable $k$), which aligns with Thm. 6.2. Yet, this leaves open exactly when practitioners should expect $k > 1$ to offer any advantages.
- The scope of the expressivity results feels somewhat limited to me. The proof of the nonequivalence of sparse resistance and biharmonic WL (Thm. 6.2) relies on constructing a single counterexample and yields little intuition about *quantitative* differences in expressivity or about structures that the PEs could capture/distinguish. While this might not be easily achievable, an ideal result would be a full characterization of the models’ *homomorphism expressivities*, as, e.g., in [1]. Also, the results do not compare MPNN expressivity with $k$-harmonic PEs to the one of graph transformers with the same relative PEs.

[1] Jingchu Gai, Yiheng Du, Bohang Zhang, Haggai Maron, Liwei Wang. *Homomorphism Expressivity of Spectral Invariant Graph Neural Networks.* ICLR 2025.

**Questions:**

- Could you clarify how you compute the $k$-harmonic distance matrix in practice and how this scales?
- Why restrict to local message passing if the $k$-harmonic matrix is dense, and how would global aggregation change the expressivity?
- Can you make any statement or hypothesis about in *which* settings $k=1$ or $k=2$ / $k > 2$ would work best?

---

> ### Author Response · Authors · 2025-11-21
>
> We thank you for your time and valuable feedback on our paper
>
> -To your point about the time complexity, the real bottleneck is not typically the time to compute the positional encoding, but rather the time to train the model. Therefore, while it would in general cost similar time to compute the positional encoding for each model (although it is cheaper for MPNNs in certain cases, see the below point), MPNNs have a significant time advantage as they take much less time to train than dense Transformers. We observe that the wall-clock training time for the equivalent transformer model is roughly an order of magnitude greater than that of the MPNN. Hence, the sparse structure make MPNNs a practical option for a majority of real-world datasets. We will add an explicit discussion of these points to the manuscript, as we agree with your characterization that these points are not immediately clear.
>
> -As we mentioned above, there are some cases where we do not need to directly compute the pseudo-inverse of the Laplacian in order to compute the k-harmonic distances on the edges; in these cases, there is an advantage to just having to compute the k-harmonic distance for the edges rather than all-pairs distances for the transformer. The biharmonic distance and effective resistance can be approximated by combining Johnson-Lindenstrauss projection and fast Laplacian of the Laplacian in nearly linear time (in the number of edges), see [1,3]. We also have a proof generalizing this speed-up to k-harmonic distances for small k; this adds a factor of k to the runtime, so approximately O(km)). However, this is a recent original result and does not yet appear in the literature or our paper. We intend to add this to a revised version of the paper.
>
> -To your point about whether transformers have the same expressive power as MPNNs, this is actually an open question we are still working on. However, it is not immediately true that transformers are generally better. The reason why they may not be better is that transformers potentially lose information when using something like the dense k-harmonic matrix as a positional encoding, as it is not clear whether these transformers are able to recover which pairs of nodes are connected by edges in the graph (or to put this formally, whether transformers+k-harmonics are more powerful than the 1-WL test). Hence, another argument in favor of MPNNs is that their structure naturally contains the information about which nodes are connected by edges, which is in addition to their improved scalability.
> For transformer comparisons, it is also worth noting that the equivalent transformer model performs the same on BREC (line 1225), and any transformer augmented with k-harmonic distances is weaker than 3-WL, as any function on the graph laplacian’s eigenvalues is weaker than 3-WL (see [2]).
>
> -The main message we sought to convey with our paper is that there is expressive granularity present in the k-harmonics both theoretically and empirically, and that this granularity is worth discussing.  As you mentioned, both the WL test and BREC would imply that the expressivity of the k-harmonics are all nearly identical. However, Theorem 6.1 and the previous research on resistance and biharmonic show that they have very different properties, so we might expect there to be datasets where the prediction is a function of one of the distances or the other.  Further, we get consistent statistically significant results showing that specific k is better for different datasets. Characterizing why different k-harmonics perform better for different datasets is open to future research, and is an exciting new direction not mentioned in the literature.
>
> --While we agree that a more quantitative result about the specific substructures that these PEs can capture is more desirable, the point of showing this difference between sparse resistance and biharmonic WL was to argue that not all k-harmonics are equal to one another, which is perhaps the conclusion a reader might make after reading Section 5. While we do construct a single counterexample, this result taken in conjunction with Theorem B.1 implies that sparse biharmonic-WL >= sparse resistance WL on all trees (by time). However, we admit that this result is only implied and not explicitly stated, which we will edit in the manuscript.
> -It is true that we do not take this approach to comparing the sparse k-harmonic WL test to the generalized k-harmonic WL test (transformers).
>
> [1] Z. Zhang, W. Xu, Y. Yi and Z. Zhang, "Fast Approximation of Coherence for Second-Order Noisy Consensus Networks," in IEEE Transactions on Cybernetics, 2020.
>
> [2] Bohang Zhang, Lingxiao Zhao, and Haggai Maron. On the expressive power of spectral invariant graph neural networks. ICML, 2024.
>
> [3] Spielman, Daniel A., and Nikhil Srivastava. "Graph sparsification by effective resistances." Proceedings of the fortieth annual ACM symposium on Theory of computing. 2008.

---

> > ### Comment · Reviewer_9Eb6 · 2025-11-26
> >
> > I thank the authors for their detailed responses and clarifications.
> >
> > **Re MPNN vs GT/complexity:** Thank you very much for the clarification, the point that MPNNs need only edge-wise k-harmonic distances (and this can be efficiently calculated) is useful context. In a way, this makes the comparison to the GT still an important baseline but less crucial as I thought before. Re efficient computation, I appreciate these arguments and the runtime observations (which would be quite important additions in my eyes), but these are not yet reflected in the current version of the paper, and the JL approximation for general k is a nontrivial result that I can’t really evaluate within this submission, solely judging from the sketch in the rebuttal message.
> >
> > **Expressivity relative to transformers:** I appreciate the clarification that the precise comparison between k-harmonic MPNNs and transformer architectures with the same PEs is currently open, and that the paper does not claim superiority of one over the other. As it stands, the paper provides a solid expressivity analysis, but does not really position k-harmonic MPNNs in the broader space of spectral transformer architectures. I see this as a limitation of scope rather than a flaw.
> >
> > **Nature of the expressivity results:** Thank you very much for your clarification; I however maintain my point that this section (which is positioned as a main contribution of the work) feels somewhat incomplete and a more structural/quantitative discussion of expressivity differences would be more desirable and provide a complete picture. This concern is also shared with reviewer BRcr.
> >
> > **Experiments and practical guidance:** The rebuttal somewhat sharpened the empirical message, but I still feel that the improvements are modest and that the paper offers limited prescriptive guidance to practitioners beyond a message like “try k=1,2 (or a learnable k) and be cautious with larger k.”
> >
> > In summary, I thank the authors for their detailed rebuttal; however, I feel like my main points haven’t changed significantly. In my opinion, the work definitely has merit, but feels incomplete/narrow in scope and, in some regards, like work in progress that has not yet reached a fully conclusive state, so I feel like it could benefit a lot from the discussed additions. For now, I will keep my score.

---

### Meta-Review · Area_Chair_Hn1N · 2025-12-24

**Summary:**

This paper studies k-harmonic distances as positional encodings for MPNNs, introducing a sparse k-WL test framework and establishing that these encodings increase expressivity beyond 1-WL but remain bounded by 3-WL. The authors show theoretical distinctions between different k values and validate empirically that optimal k is dataset-dependent. The reviewers acknowledge that studying the expressivity of k-harmonic distances is a meaningful contribution, with technically sound proofs and a clean theoretical framework. However, reviewers express concerns about the limited significance of the theoretical results: the expressivity analysis relies on isolated counterexamples rather than structural characterizations, the experimental improvements are modest and dataset-dependent.

**Reviewer Concerns:**

The authors' rebuttal addressed computational complexity concerns by clarifying that MPNNs only need edge-wise distances (not full dense matrices) and pointing to nearly-linear time approximations. The comparison to transformers remains unresolved—the authors acknowledge this is an open question. The core concern about limited structural insight from the expressivity results (raised by all reviewers) was only partially addressed.

**Reviewer Scores:**

Reviewers would likely maintain their score of 2,4,6 as main concerns remain only partially resolved

---

### Decision · Program_Chairs · 2026-01-26

Reject